# Systematic discovery of linear binding motifs targeting an ancient protein interaction surface on MAP kinases

András Zeke[1], Tomas Bastys[2,3], Anita Alexa[1], Ágnes Garai[1], Bálint Mészáros[4], Klára Kirsch[1], Zsuzsanna Dosztányi[5], Olga V Kalinina[2] & Attila Reményi[1,*]

## Abstract

Mitogen-activated protein kinases (MAPK) are broadly used regulators of cellular signaling. However, how these enzymes can be involved in such a broad spectrum of physiological functions is not understood. Systematic discovery of MAPK networks both experimentally and *in silico* has been hindered because MAPKs bind to other proteins with low affinity and mostly in less-characterized disordered regions. We used a structurally consistent model on kinase-docking motif interactions to facilitate the discovery of short functional sites in the structurally flexible and functionally under-explored part of the human proteome and applied experimental tools specifically tailored to detect low-affinity protein–protein interactions for their validation *in vitro* and in cell-based assays. The combined computational and experimental approach enabled the identification of many novel MAPK-docking motifs that were elusive for other large-scale protein–protein interaction screens. The analysis produced an extensive list of independently evolved linear binding motifs from a functionally diverse set of proteins. These all target, with characteristic binding specificity, an ancient protein interaction surface on evolutionarily related but physiologically clearly distinct three MAPKs (JNK, ERK, and p38). This inventory of human protein kinase binding sites was compared with that of other organisms to examine how kinase-mediated partnerships evolved over time. The analysis suggests that most human MAPK-binding motifs are surprisingly new evolutionarily inventions and newly found links highlight (previously hidden) roles of MAPKs. We propose that short MAPK-binding stretches are created in disordered protein segments through a variety of ways and they represent a major resource for ancient signaling enzymes to acquire new regulatory roles.

**Keywords** cellular signaling; linear motif; MAP kinase; protein–protein interaction

**Subject Categories** Computational Biology; Signal Transduction; Structural Biology
**Mol Syst Biol.** (2015) 11: 837

## Introduction

Protein–protein interactions influence all aspects of cellular life and the most direct mechanism through which proteins can influence each other is by physical interaction. This brings them into proximity to exert control on activity or to create opportunities for posttranslational modification. Protein–protein associations often involve so-called linear binding motifs which are short (5–20 amino acid long) protein regions lacking autonomous tertiary structure. These functional sites reside in intrinsically disordered protein regions and adopt stable conformation only upon binding. Currently, we can only guess how abundant linear motif-based interactions are; nevertheless, it was recently estimated that there are ~100,000 linear binding motifs targeting dedicated protein surfaces in the human proteome (Tompa *et al*, 2014). As an example relevant to cellular signaling, mitogen-activated protein kinases (MAPKs) are prototypical enzymes that depend on short segments from partner proteins and on their dedicated protein–protein interaction hot spots. They mainly recognize their substrates not with the catalytic site but with auxiliary docking surfaces on their kinase domains (Tanoue *et al*, 2000; Biondi & Nebreda, 2003). The most important of these docking sites consists of a hydrophobic docking groove and the negatively charged CD (common docking) region (Chang *et al*, 2002) (Fig 1A). Together, they can bind the so-called D(ocking)-motifs of the target proteins. D-motifs are short linear motifs ranging from 7 to 18 amino acids in length and are typically found in intrinsically disordered segments—potentially far away from target phosphorylation sites (Garai *et al*, 2012). Such docking

1  Lendület Protein Interaction Group, Institute of Enzymology, Research Center for Natural Sciences, Hungarian Academy of Sciences, Budapest, Hungary
2  Max Planck Institute for Informatics, Saarbrücken, Germany
3  Graduate School of Computer Science, Saarland University, Saarbrücken, Germany
4  Institute of Enzymology, Research Center for Natural Sciences, Hungarian Academy of Sciences, Budapest, Hungary
5  MTA-ELTE Lendület Bioinformatics Research Group, Department of Biochemistry, Eötvös Loránd University, Budapest, Hungary
   *Corresponding author. Tel: +36 1 3826613; E-mail: remenyi.attila@ttk.mta.hu

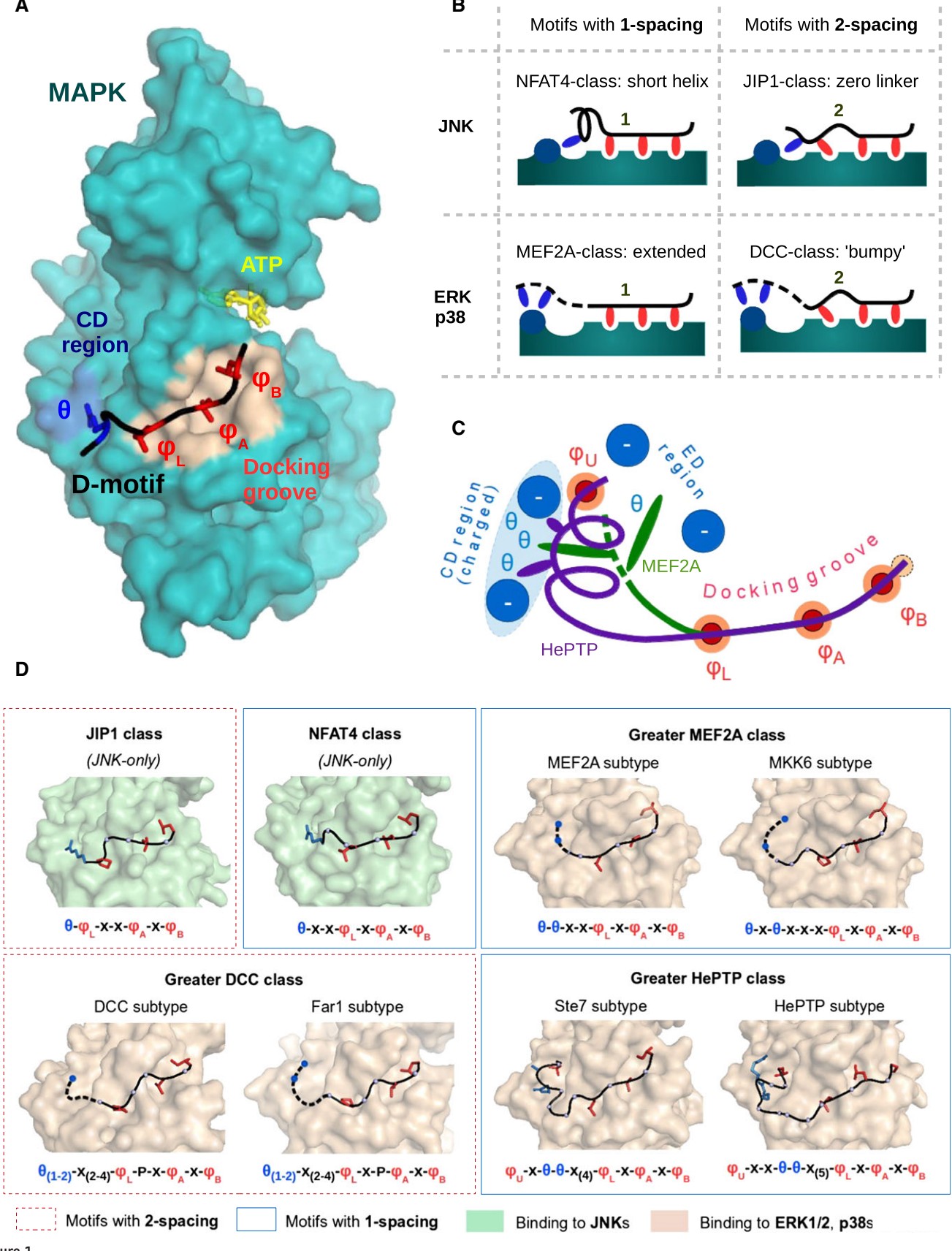

**Figure 1.**

**Figure 1.  Structural classification of MAPK-docking motifs.**

A   The MAPK-docking groove comprises two distinct regions: the common docking (CD) and the hydrophobic region. These are colored in blue and light brown, respectively, and are shown on the JNK1 surface from the JNK1-NFAT4 protein–peptide complex crystal structure (Garai *et al*, 2012). (The CD groove is extended by the ED region, extra negatively charged residues for ERK and p38; see (C); Tanoue *et al*, 2001.)

B   Different binding modes of D-motifs. The hydrophobic docking groove binds three hydrophobic amino acids in a row, while admitting two different spacing schemes. At the same time, θ to φ linker length determines the MAPK specificity of a given motif. These two features can combine freely with each other, resulting in the four basic arrangements shown here.

C   Distinct binding conformations at the CD groove. N-termini of longer D-motifs are variable and ERK2- or p38α-binding peptides may take a variety of conformations —ranging from fully linear (e.g., MEF2A, green) to highly alpha-helical (e.g., HePTP, magenta).

D   Structural heterogeneity of D-motifs. The combinations of the three variable features yield structurally well-defined, distinct classes of D-motifs. Many of these models also define separate types of linear motifs, but their consensus sequences are not always exclusive. JNK kinases only admit two major types of motifs, the NFAT4 class (1-spacing, short linker) and the JIP1 class (2-spacing, short linker). On the other hand, known ERK1/2 and p38 binder peptides may belong to the greater MEF2A class (1-spacing, longer linker, linear end), the greater HePTP class (1-spacing, longer linker, helical end), or the greater DCC class (2-spacing, longer linker, linear end). A sixth class of ERK or p38 interactors is theoretically also possible (2-spacing, longer linker, helical end), but this combination can only be observed in long reverse (revD) motifs (Garai *et al*, 2012), and no classical motif of this type is known up to date. Subtypes and other variants within a given greater class are also featured wherever applicable. These are shown based on structures of MAPK-docking motif complexes. Dashed lines indicate N-terminal peptide regions that are usually not visible in the crystal structures, albeit indispensable for binding. Consensus motif of each subtype is shown below, where $\varphi_U$, $\varphi_L$, $\varphi_A$, and $\varphi_B$ letters denote positions that are filled by hydrophobic amino acids—L, A, and B refer to the lower pocket, and pockets A and B, respectively—while the θ positions are positively charged (Arg or Lys) while letter "x" denotes arbitrary amino acids.

elements are not only restricted to substrates: They are also found in MAPK-activating kinases (MAP2Ks), in MAPK-inactivating phosphatases (MKPs), and in a variety of scaffold proteins. While extracellularly regulated kinases (ERK1-2), c-Jun N-terminal kinases (JNK1-3), and the 38-kDa protein kinases (p38α-δ) control diverse physiological processes, they phosphorylate most of their substrates at Ser-Pro or Thr-Pro (S/TP) sequence motifs. Naturally, this weak consensus provided by their catalytic site is insufficient for selective target recognition, and additional linear binding motifs provide specificity (Johnson & Lapadat, 2002; Bardwell, 2006). Therefore, the MAPK D-motif protein–protein interaction system is an ideal test bed for linear binding motif discovery.

Several previous attempts were aimed at predicting MAPK-binding proteins from full proteomes by using a generic consensus of D-motifs as it had been established more than a decade ago (Sharrocks *et al*, 2000). This consensus was derived from an observation that D-motifs almost always include at least a single positively charged residue (termed the θ position: arginine or lysine) and a series of alternating hydrophobic residues (φ positions: frequently leucine), connected by a linker of a variable length (Dinkel *et al*, 2014). But despite the use of extensive multiple alignments and sophisticated algorithms, predictions had only low success rates and large-scale assessment of predicted hits was not performed (Whisenant *et al*, 2010; Garai *et al*, 2012; Gordon *et al*, 2013). Regarding experimental MAPK network discovery, ERK2 has been the most widely explored. For example, several different methods were utilized to identify ERK2 substrates by large-scale phosphoproteomics (Kosako *et al*, 2009; Carlson *et al*, 2011; Courcelles *et al*, 2013). Unfortunately, pairwise overlaps between the lists of substrates are low across studies (e.g., around ~10%), with not a single overlap between five different studies that aimed to find ERK2-phosphorylated substrates (Courcelles *et al*, 2013), suggesting great dependence on the experimental conditions used. It was noted that D-motif-like sequences are enriched in experimentally detected ERK2 substrates (Carlson *et al*, 2011), yet detection or verification of direct physical association was not performed. In addition, studies that used a high-throughput approach to identify partners of JNK1 (Chen *et al*, 2014) or p38α (Bandyopadhyay *et al*, 2010) based on direct physical interaction resulted in low number of hits. Thus, it is likely that a protein–protein interaction-based MAPK network

discovery could greatly benefit from a target tailored approach, which takes into account—and possibly capitalizes on—specific biochemical and biophysical knowledge already available on known MAPK–partner protein interactions.

In recent years, the number of experimentally determined MAPK–partner protein complex structures increased considerably (Garai *et al*, 2012). This development made it possible to amend the definition of the underlying sequence motifs and it became clear that D-motifs encompass multiple classes of similarly built, but structurally distinct linear motifs (similar to SH3- or PDZ-domain-binding sequences, for example) (Lim *et al*, 1994; Tonikian *et al*, 2008). In the current study, we show that by building a strategy that can handle this conformational diversity in binding, and using structural compatibility with specific interaction surface topography as an additional criterion for prediction, the identification of novel D-motifs can be dramatically improved. This analysis in combination with tailored experimental techniques for the validation of low-affinity (1–20 μM) protein–protein interactions produced unique, molecular-level insight into physiological roles and evolution of MAPK-based protein networks.

## Results

### Structure-guided prediction of MAPK-binding linear motifs

MAPK–D-peptide complex structures revealed distinct D-motif binding modes in the MAPK-docking groove (Fig 1). For example, D-motifs from the JNK-binding scaffold protein JIP1 and from the JNK-regulated transcription factor NFAT4 bind to the same docking surface differently (Fig 1A and B) (Heo *et al*, 2004; Garai *et al*, 2012; Laughlin *et al*, 2012). Similarly, ERK- and p38-binding D-motifs may also be structurally distinct; nonetheless, D-motifs could be described with a common loosely defined consensus [$\theta_{1,2}$-x$_{(0-5)}$-φL-x$_{(1,2)}$-φA-x-φB; φL, φA, and φB denote positions that are typically filled by hydrophobic amino acids—L, A, and B refer to the lower pocket, and pockets A and B, respectively—while the θ denotes positively charged (arginine or lysine) and "x" denotes any amino acid]. However, the rules are much stricter for sequences that are compatible with a given MAPK-docking surface in a given binding

mode. Interestingly, D-motifs and their binding modes may be conserved from yeast to human as the docking surface is ancient and well conserved across all eukaryotes (Reményi *et al*, 2005; Grewal *et al*, 2006).

Because the CD region of ERK and p38 is wider compared to that of JNK, the N-termini of motifs binding to the two former kinases have larger conformational freedom (Fig 1C) (Garai *et al*, 2012). These can be classified as MEF2A- and DCC-type motifs named after the proteins in which they were first identified. Some motifs with longer intervening regions also exists (HePTP) (Zhou *et al*, 2006). Interestingly, the typical helical conformation at the N-terminus of HePTP-type docking motif is also characteristic to some MAPK inter-actors from yeast (Ste7) and peptides with such motifs are known to bind human ERK2 with high affinity (Fernandes *et al*, 2007). There-fore, we also set up a hypothetical subclass of Ste7-type motifs, hith-erto unknown in humans (Fig 1D).

Simple pattern matching of motifs normally produces a large number of false positives, because motif-matching sequences may occur simply by chance. In order to drastically reduce the number of false hits, an *in silico* filtering procedure was implemented to search for putative linear motifs (Fig 2). The first step was to screen for motifs in regions with intrinsic disorder but with propensity for disorder-to-order transition (ANCHOR) (Dosztányi *et al*, 2009; Mészáros *et al*, 2009). This procedure was used in order to elimi-nate those consensus motif occurrences that would either be buried in folded domains or permanently locked in an unfavorable confor-mation. Importantly, it also removed initial hits with an inappropri-ate amino acid composition, not being able to adopt a stable structure upon binding to a protein surface. Motifs were then filtered for MAPK accessibility: Motifs that were predicted to lie in extracellular protein segments or in other, kinase-inaccessible compartments (e.g., ER, Golgi) were discarded. In addition, an auxiliary check was performed against structured Pfam domains. This was applied to remove all spurious motifs in ordered regions which had been retained after ANCHOR filtering. Since almost all the known motifs passed these filters (45 out of 47, with enrichment ratios between 4.1 to 6.6 depending on motif type), we concluded that the dataset was of sufficient quality for further testing.

The classification of D-motifs based on a coherent structural model enabled us to make use of structure-based scoring. As a motif occurrence could always be unambiguously matched with its corresponding MAPK-docking peptide structural model, we used FoldX, which had been validated on protein–peptide complexes, to estimate the change of the protein–peptide binding energy with respect to the energy of an experimentally resolved complex (Appendix Table S1) (Schymkowitz *et al*, 2005). This allowed the scoring of motifs according to their structural compatibility to the MAPK-docking groove. FoldX-derived binding energy estimates were also used as a guide when motifs were being chosen for later experimental screening.

### Experimental screening

After completion of initial lists, we chose a number of candidate proteins from each motif type to test. Expression of full-length human proteins of large size (> 1,000 amino acids) in recombinant systems can be a limiting step in experimental validation; therefore, first we opted for a fragment-based approach. Former experiments

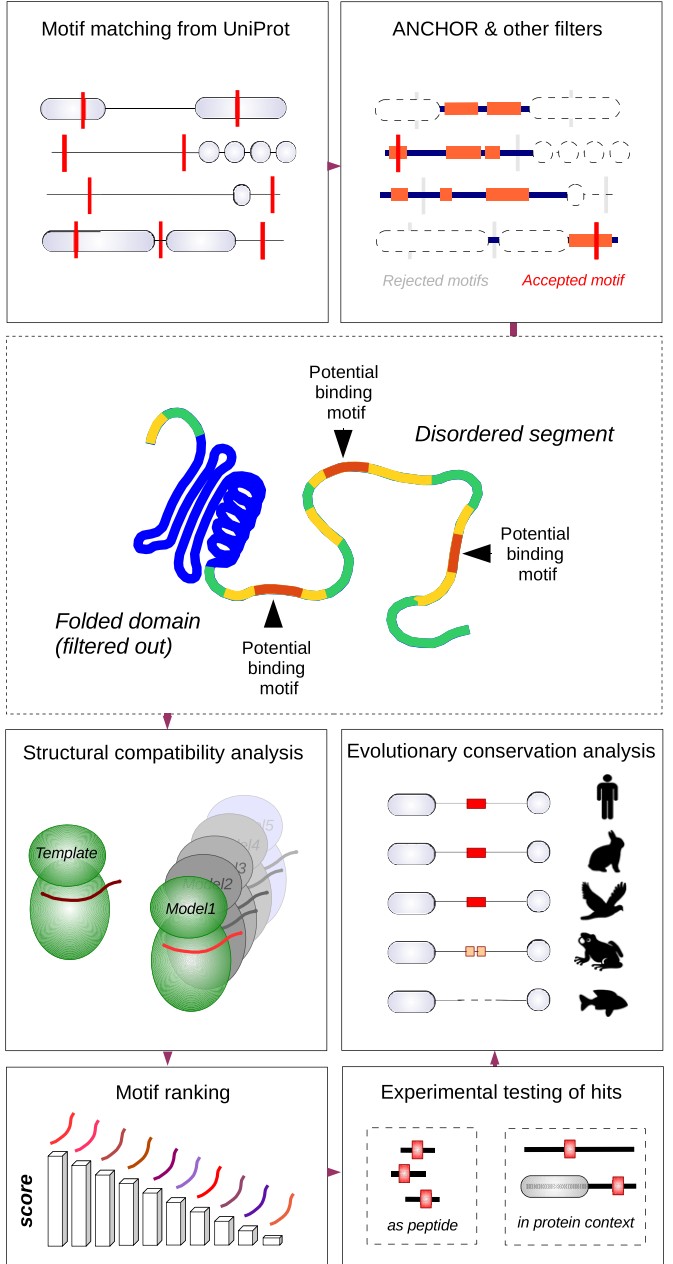

**Figure 2.  Motif finding work flow.**

To find novel MAPK-docking motifs, the primary motif-matching step (on UniProt KB sequences) was followed by a series of filters. Valid motifs had to pass through an ANCHOR filter, a localization filter (combined from SignalP and Phobius) and an auxiliary Pfam filter, in order to be scored by FoldX homology models. ANCHOR (middle panel) had the most important role in selecting segments that can potentially act as linear motifs, while FoldX gave motif-specific binding energy estimates (see Source data). Predicted motifs were subsequently tested as short fragments and (in selected cases) as full-length proteins. Finally, an automated evolutionary analysis was performed to give information on motif conservation trends.

Source data is available online for this figure.

showed that simple binding assays (such as pull-downs with recom-binant D-motif-containing proteins or immobilized solid-phase peptide arrays) lack the robustness to reliably detect low-affinity

(1–20 µM) protein–peptide interactions. Therefore, we developed a different assay which was based on substrate phosphorylation enhancement on a solid-phase support (Fig 3A). As the majority of known D-motif-containing proteins are MAPK substrates, this adequately captures the original function of these motifs. An artificial substrate was constructed containing the D-motifs as well as the Thr71 phosphorylation site from ATF2, which is a well-known MAPK target site (Livingstone *et al*, 1995) (Fig 3B). As linkers and substrate sites in the recombinant proteins were identical, the "docking efficiency" of the given motifs could be directly compared to each other. Phosphorylation of this reporter solely depended on the presence or absence of specific docking motifs, and phosphorylation of the target site was low without a functional D-motif (Fig 3C and D).

In the final panels, we included 70 different constructs: 63 of these were directly selected from the lists ranked by the predicted binding energy (Fig EV1 and Appendix Fig S1). We also included additional seven motifs based on sequence similarity to known motifs. This was done in order to test whether some other similar motifs not conforming to the formerly defined sequence patterns have the capacity to bind MAPKs. Out of 70, a total of 52 motifs were found to interact with at least one of the three MAPKs (ERK2, JNK1, or p38α). In particular, we were able to detect several novel interactors based on the JIP1, NFAT4, MEF2A, MKK6, and DCC models. As for our hypothetic Ste7 model, we also found a novel hit: a motif from RHDF1 that is also found in the related RHDF2 protein. Such a high number of hits suggest that D-motifs are in fact quite widespread in the human proteome (Table 1, Fig EV2, and Appendix Fig S2).

To show that the phosphorylation enhancements were indeed due to the presence of canonical MAPK D-motifs binding into the MAPK-docking groove, a set of 16 chemically synthesized peptides were titrated against fluorescently labeled control peptides known to bind at the MAPK-docking groove (Fig EV3 and Appendix Fig S3). In addition to confirming binding in the MAPK-docking groove, the dissociation constant (Kd) of unlabeled test peptides could also be calculated.

Binding affinities obtained through this fluorescence polarization (FP) based *in vitro* assay also allowed us to examine the specificity profiles of D-motifs. The tested peptides could be clustered into two groups based on their sequences and affinities. Similar to earlier results, these experiments confirmed the strong correlation between the ability of a given motif to bind ERK2 and p38α. Binding results also reflected the fundamental lack of correlation between ERK2/p38α and JNK1 association (Garai *et al*, 2012). These observations did agree well with phosphorylation enhancement results from dot blots. There was no positive correlation between the profiles of the JNK1 vs. p38α or the JNK1 vs. ERK2 pairs (Pearson's $r = 0.003$ and $r = -0.280$, respectively). At the same time, a modest correlation was observed between ERK2 and p38α ($r = 0.680$). This MAPK profiling confirmed our structural models. Practically, no strong JNK1-binding motif was found from other than the JIP1- or NFAT4-type classes. Most novel p38α interactors, on the other hand, belonged to the MEF2A, MKK6, or DCC types as expected.

To test whether docking motifs were also functional in their native protein context, we set up a bimolecular fluorescent protein fragment complementation (BiFC)-based cellular assay (Fig 4A). In this series of experiments, one fragment of YFP was fused to either ERK2, JNK1, or p38α. The other fragment was joined to the test protein, and fluorescence intensity was measured in live HEK293 cells. BiFC signals were always compared to the results obtained with the same construct but lacking the docking motif. Although the transgenes were overexpressed, comparison of the BiFC signal between wild-type and D-motif-mutated MAPK partner proteins could be reliably used to infer D-motif-dependent interactions within cells.

Well-known MAPK partners, such as MKK1, JIP1, or MKK6, displayed a pattern consistent with the specificity of their D-motifs. Such interactions are also greatly diminished or abrogated after the loss of the docking motif, similar to novel MAPK partners (Fig 4B). For example, the MEF2A-type motif-bearing AMP-activated protein kinase subunit γ2 (AAKG2) interacted with p38α (and ERK2), but not with JNK1. Interestingly, AAKG2 is known to have multiple shorter isoforms and it uses alternative initiation codons. One such variant (isoform C) is only 44 amino acids shorter. This natural deletion mutant lacking the N-terminal (MEF2A-type) docking motif showed a greatly reduced level of fluorescence for both partners. The differences in fluorescence were readily visible on cells under a fluorescent microscope (Fig 4C). All intensities, as well as their reduction in the mutants, were also comparable to those observed in control experiments (Fig 4D). These results are well in line with *in silico* predictions and *in vitro* fragment-based experiments. To this end, we tested six predicted motifs (AAKG2, MKP5, RHDF1, KSR2, DCX, APBA2), and one non-binder based on results of dot-blot arrays (FAM122A) was also included (Fig 4B and Appendix Fig S4). Results of this cell-based approach were consistent with the structural models as well as with the results of *in vitro* experiments.

## D-motif-based MAPK interactomes

Next, we utilized the experimentally validated new D-motifs to further improve our initial structural models. Evolutionary conservation analysis on motifs was also used to examine sequence conservation or diversity per each position (Fig 5A). Once the consensus sequences were improved, we set out to build a sequence-based method to enable direct search for MAPK-interacting proteins from the human proteome. Position-specific scoring matrices (PSSMs) were constructed from full sets of evolutionarily related docking motifs. PSSM-based profiles have been used in multiple databases for encoding information about sequence profiles, in the search for proteins with distant similarity, and they were recently used for detecting MAPK target phosphorylation sites (Schäffer *et al*, 2001; Hulo *et al*, 2004; Gordon *et al*, 2013).

High area under curve (AUC) values of the receiver operation characteristic (ROC) curve from a fivefold cross-validation with validated binders and simulated non-binders (see Materials and Methods) imply an adequate coverage of motifs in the JIP1, NFAT4, and greater MEF2A classes: 0.98, 0.94, and 0.97, respectively (Table EV1). The correlation with the original, FoldX-based rankings was modest, but clearly present in the case of JIP1-type ($r = -0.62$) and NFAT4-type ($r = -0.59$) motifs (where best correlation is $-1$, due to the negative energy scale). It was lower for the DCC ($r = -0.40$), MEF2A ($r = -0.30$), and MKK6 ($r = -0.26$) models, as somewhat expected, since the structural templates of these were incomplete (as the structures of the charged N-termini of some D-motifs are not known). Unfortunately, the lack of sufficiently diverse hits among DCC and HePTP-type motifs made PSSM construction impractical. A PSSM was still built for the greater DCC

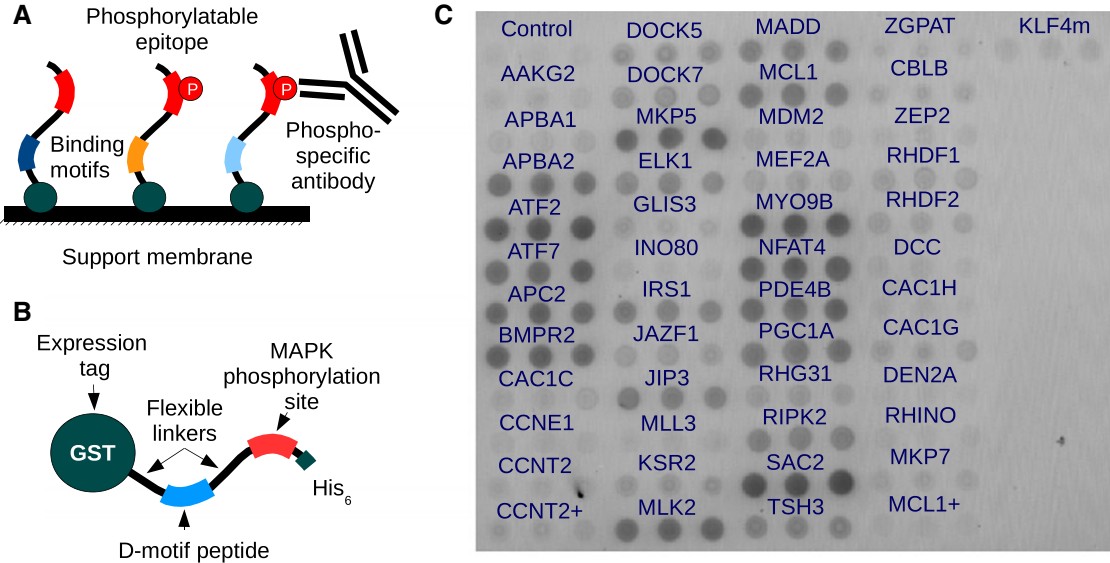

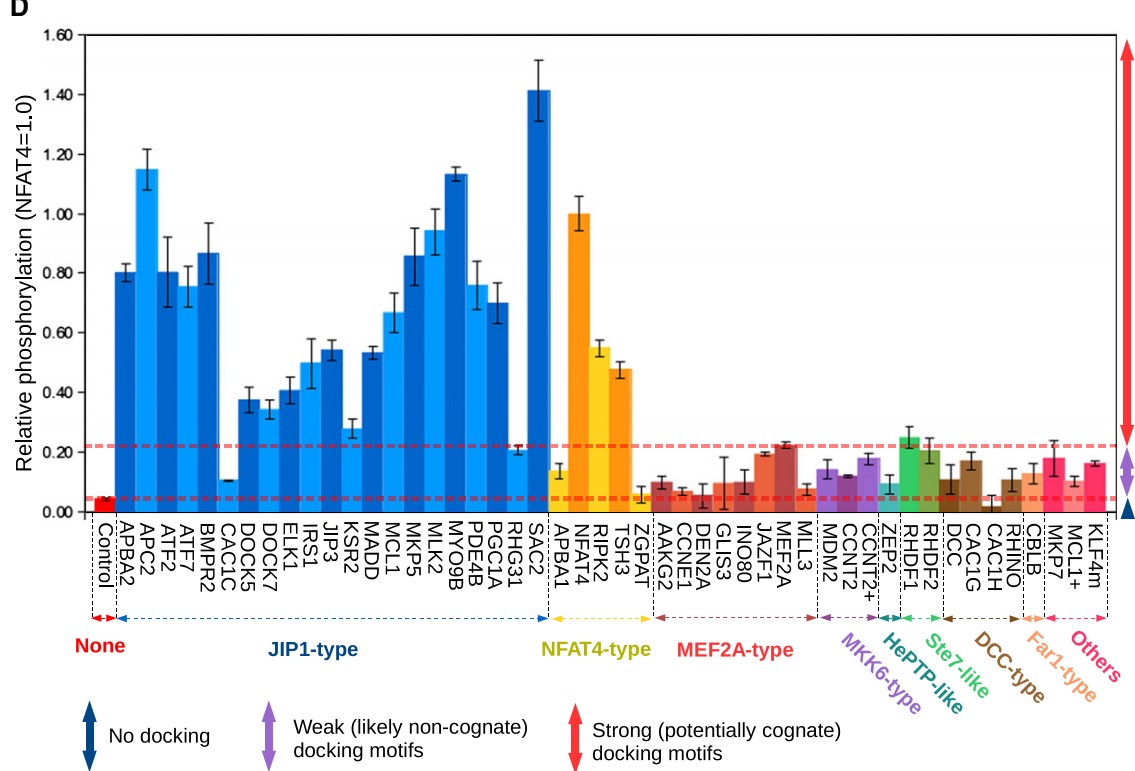

**Figure 3.    Dot-blot arrays of D-motifs.**

A    The principle of the phosphorylation enhancement dot-blot arrays. Protein constructs are immobilized onto a solid-phase support where phosphorylation takes place. Afterward, the phosphorylated epitopes are detected through standard Western blot procedures using a phosphorylation-sensitive antibody.

B    The schematic structure of the artificial substrates utilized in the dot-blot arrays. All constructs share the same tags, substrate sites, and linkers: Only the docking motif-containing fragments differ.

C    A sample dot-blot array for detecting JNK-binding docking motifs. This specific array contains 48 of the 70 motifs tested in total, and was incubated against activated JNK1.

D    Quantitative analysis of the sample dot-blot array. All intensities are relative to that of the NFAT4 motif (positive control), error bars were derived from three parallel samples on the same membrane and show standard deviation from the mean ($N$ = 3). "+" denotes additional, non-overlapping motifs tested from the same protein. "m" refers to murine (non-human) sequence. (The corresponding ERK2 and p38α 48 motif arrays and the 70 motif arrays for all three MAPKs are shown on Appendix Fig S1 or on Fig EV1).

**Table 1.  Validated sequences grouped by D-Motif class.**

| Greater MEF2A class (phosphorylation by p38α) | | | Greater DCC class (phosphorylation by ERK2) | | Greater HePTP class (phosphorylation by ERK2) | |
|---|---|---|---|---|---|---|
| **MEF2A-type** | **MKK6-type** | **Misc.** | **DCC-type** | **Far1-type** | **Ste7-like** | **HePTP-like** |
| AAKG2/PRKAG2 (28–37) | CCNT2 (498–509) | AMPD1 (109–120) | **DCC (1,144–1,155)** | CBLB (489–500) | RHDF1/iRhom1 (11–24) | ZEP1/HIVEP1 (1,422–1,438) |
| JAZF1 (77–86) | GAB3 (363–374) | AMPD3 (79–90) | CACNA1G (1,030–1,041) | ELMSAN1 (601–612) | RHDF2/iRhom2 (18–31) | |
| INO80[a] (1,318–1,327) | INO80[a] (1,316–1,327) | | | TRERF1 (653–664) | | |
| **MEF2A (268–277)** | KSR2 (330–341) | | | GAB1 (526–536) | | |
| KLF3 (88–97) | KMT2C/MLL3[a] (1,195–1,206) | | | | | |
| KMT2C/MLL3[a] (1,197–1,206) | | | | | | |
| RIPK2[a] (326–335) | | | | | | |
| TSHZ3[a] (321–330) | | | | | | |
| **JIP1 class (phosphorylation by JNK1)** | APBA2/MINT2 (279–285) | ATF2 (164–170) | **ATF7 (162–168)** | APC2 (962–968) | **BMPR2 (753–759)** | DOCK5 (1,762–1,768) |
| | DOCK7 (884–890) | DUSP10/MKP5 (18–24) | **ELK1 (314–320)** | **IRS1 (856–862)** | **JIP3 (203–209)** | M3K10/MLK2 (876–882) |
| | MADD (809–815) | MCL1 (136–142) | **MYO9B (1,249–1,255)** | PDE4B (72–78) | PRGC1/PPARGC1A (253–259) | SAC2/INSPP5 (1,009–1,015) |
| **NFAT4 class (phosphorylation by JNK1)** | **AKAP6/mAKAP (433–440)** | CCSER1 (573–580) | DYH12/DNAH12 (12–19) | FMN1 (672–679) | FHOD3 (506–513) | **JUND (46–53)** |
| | KANK2 (244–251) | M3K10/MEKK1 (1,077–1,084) | **MABP1 (1,292–1,299)** | **NFATC3/NFAT4 (145–152)** | RIPK2[a] (327–334) | TSHZ3[a] (322–329) |

The table lists motifs that tested as positive ("hits") in the dot-blot arrays and it shows the most commonly used names of proteins (when necessary, two variants), together with the position of the core motif—based on the reference isoform featured in UniProt. Names in bold type denote previously known docking motifs, while the names in normal type indicate novel interactors.
[a]Denotes motifs that appear under more than one class as they satisfy multiple consensus sequences.

class, but only to compare it to the other three. This comparison showed that in several cases, positional preferences could be explained on a structural basis (Fig 5B and C).

The structurally consistent PSSM-based search method offered us a unique glimpse into the human MAPK interactome, albeit limited to D-motif-containing proteins. As it included a rather large number of proteins that have little or no formal Gene Ontology (GO) annotation, we decided to annotate the best 100 hits manually, based on UniProt labels, domain composition, and literature (Table EV2). Out of the three classes examined, the JIP1 type had by far the highest number of validated hits. Thus, the predictions for this class were deemed most reliable, shedding some light on the interactome of JNK1 (Fig 6). Among the less surprising categories discovered were the MAPK pathway components themselves (especially at the MAPK kinase kinase [MAP3K] level, as potential feedback elements), several transcription factors and other gene expression regulatory systems, or various ubiquitin ligases. A considerable number of experimentally tested or predicted JNK-interacting proteins have preferentially or exclusively neuronal functions. We predict that the axons, nerve terminals, and dendrites—especially in synapses—

contain a high number of specialized JNK-interacting proteins, as do developing neuroblasts and their axonal growth cones (Appendix Fig S5A). Interestingly, the majority of JNK-associating proteins (both experimentally validated and predicted) seem to be involved in cytoskeletal regulation. We encountered numerous actin-binding or microtubule-binding proteins, molecular motors as well as small G protein partners. Docking motifs were even found on proteins localized to centrosomes, basal bodies, or those involved in the formation of primary cilia. Several other high-scoring hits suggest that JNK is intimately involved in the regulation of endo- and exocytosis.

The presence of insulin signaling pathway components in the lists may also explain many previous observations on the causative role of JNK in type II diabetes. This kinase is involved in pathways over-activated by cytokines derived from adipose tissue. JNK1 knockout mice are also known to be resistant to type II diabetes induced by obesity (Hirosumi *et al*, 2002; Sabio *et al*, 2010). Proteins bearing JIP1-type docking motifs (e.g., MADD, IRS1, PGC1A) are located in critical points of networks responsible for insulin signaling, and these are the same pathways that are also targeted by most anti-

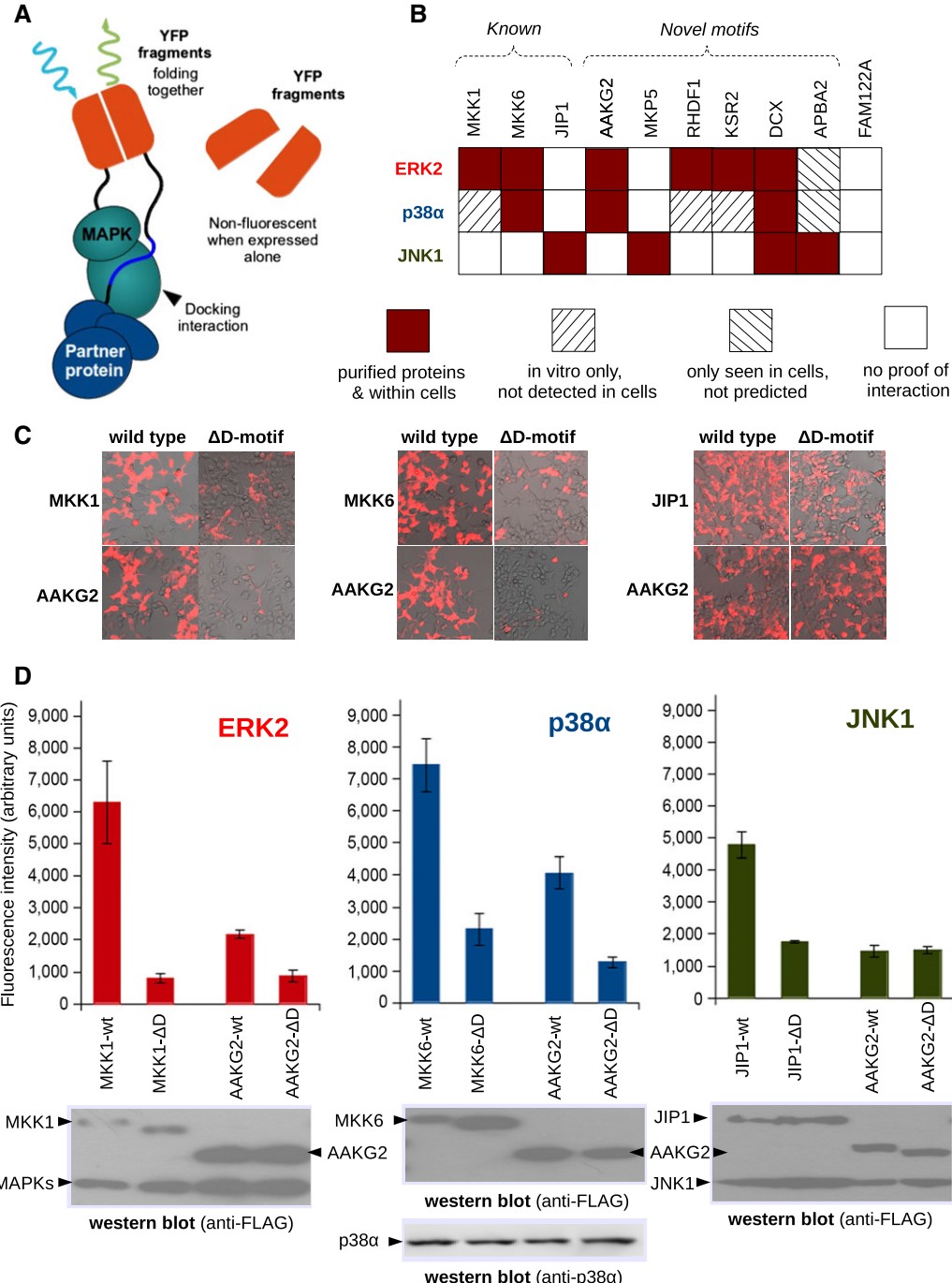

**Figure 4.   Bimolecular fragment complementation (BiFC) experiments.**

A   The principle of YFP fragment complementation driven by MAPK–partner protein interaction. The weak and transient interactions between MAPKs and its binding partners still lead to well-detectable signals.

B   Summary of the BiFC experiments. In addition to successfully testing six novel docking motif-dependent interactions, three positive controls (MKK1, MKK6, and JIP1) and an extra negative control (FAM122A) were also introduced into this analysis. Red squares indicate positive BiFC results (which were mostly directly predictable from fragment-based experiments). However, some interactions suggested by dot-blot experiments and/or FP titrations were not seen in BiFC (lined squares). These were possibly too weak or absent in the cellular context.

C   Bright-field image of transiently co-transfected HEK293 cells overlaid with the fluorescence image. Although expression levels and complementation efficiency vary between cells, ablation of D-motifs results in robust fluorescence intensity changes for known MAPK–partner protein pairs (MKK1-ERK2, MKK6-p38α, and JNK1-JIP1, from left to right, upper panels) similar to a novel MAPK partner (AAKG2, lower panels).

D   Results of fluorescence measurements on co-transfected cell populations with positive controls and for AAKG2. (Error bars show standard deviations from the mean, *N* = 6). Similar expression levels of FLAG-tagged proteins (wild-type or D-motif lacking versions of known or putative MAPK partners) or MAPKs were confirmed by Western blotting using anti-FLAG (ERK2 and JNK1) or anti-p38α antibodies. Further BiFC results are shown on Appendix Fig S4.

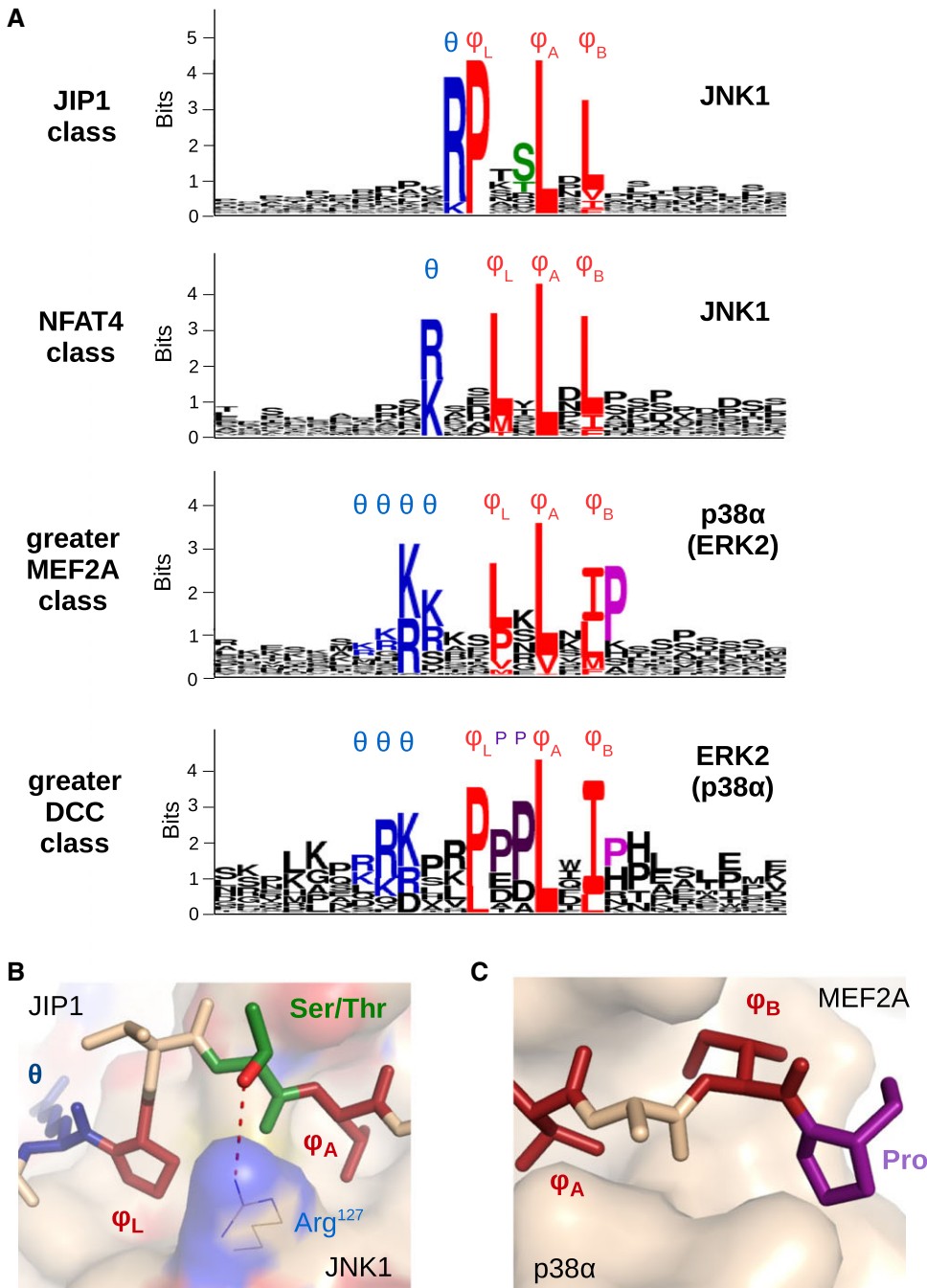

**Figure 5.  Position-specific scoring matrix (PSSM) logos.**

A  Sequence logos generated from the evolutionarily weighted PSSMs. These logos were generated for the four classes for which an adequate number of examples was available. In the core motif, the positively charged θ positions are colored blue, while the three φ hydrophobic contact points are red. The JIP1 class and the NFAT4 class were built from motifs binding to JNK1; therefore, they are directly comparable. The greater MEF2A class includes p38α-binding motifs (with a minority also binding to ERK2). At the same time, the greater DCC class contains motifs primarily binding to ERK2 (with many of its members also associating with p38α). The logo of the greater DCC class was built based on only 6 independently evolved proteins (compared to 21 for JIP1, 15 for NFAT4, and 15 for greater MEF2A classes, respectively). As the JIP1-type PSSM contains the highest number of independent examples, it is considered the most reliable, while the DCC is the most coarse.

B  Positional amino acid preferences in the PSSM matrix can be explained on structural grounds even at highly variable intervening regions between core motif positions: The JIP1-type motifs frequently have serine or threonine in the position immediately preceding φA (colored in green on the logo). From the structure of the JIP1–JNK1 complex (Heo *et al*, 2004), it is clear that this amino acid has the ability to form a hydrogen bond with the underlying arginine side chain of JNK1.

C  The greater MEF2A—and to a lesser extent, greater DCC—motif classes show a clear preference for proline after φB (colored in light magenta). The panel on the MEF2A-p38α complex shows that this proline can form an additional hydrophobic interaction toward the surface of p38α (Chang *et al*, 2002). Thus, the reason for this phenomenon is different from the preference for prolines in the greater DCC class (dark magenta on the logos) where the latter are required to favor a type II polyproline helix (Ma *et al*, 2010).

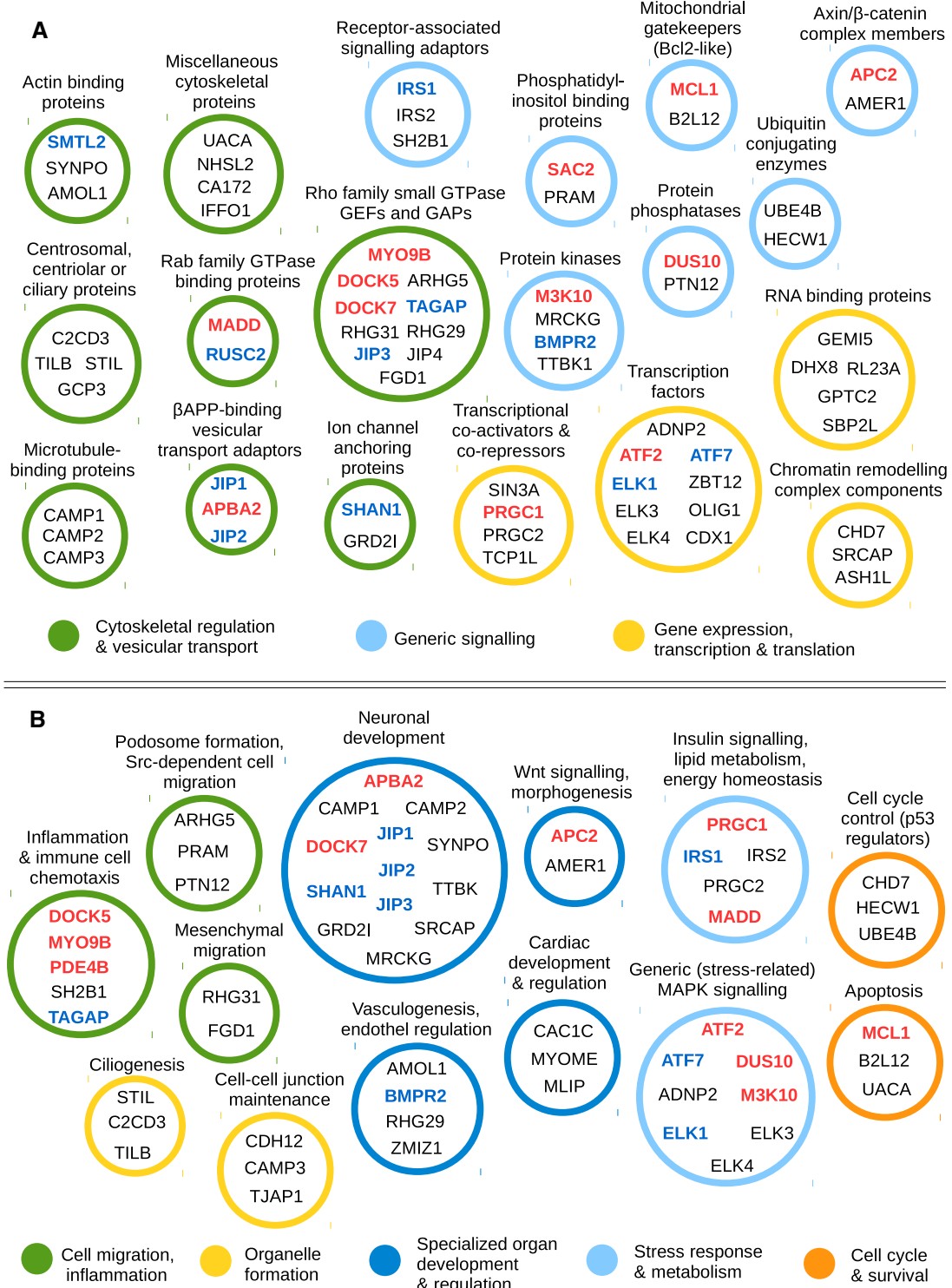

**Figure 6.  JNK interactome based on the presence of JIP1-type motifs.**

A   Low-level functional classification of JIP1-type motif-bearing proteins. This analysis reveals that cytoskeletal regulation is at least as important aspect of JNK1 signaling as control of gene expression. The best 100 hits on the JIP1-type motif list contain many proteins whose docking motif is already known (blue letters), or was validated in our dot-blot essays (red letters). Only categories that contain more than a single protein are shown.

B   High-level functional classification of the best-ranking 100 JIP1-type motif. The analysis reveals several major functions associated with JNK kinases. The role of JNK in inflammation, neuronal development, metabolic regulation, or apoptosis is already known from cell-based experiments. Motif search results, in addition to providing the mechanistic basis for these regulatory processes, also suggest novel functions. The full functional clustering of JIP1-type D-motifs along with NFAT4- and MEF2A-type motifs is shown in Table EV2.

diabetic pharmaceuticals (Lee *et al*, 2003; Finck & Kelly, 2006; Olson *et al*, 2008; Li *et al*, 2014) (Appendix Fig S5B).

Experiments with greater MEF2A-type motif-bearing proteins provided another interesting observation. The occurrence of such motifs in proteins (KSR2, AAKG2, and AMPD1,3) with an important regulatory role in the adenosine monophosphate-activated protein kinase (AMPK) pathway implies that this system connects to the p38 and/or the ERK1/2 pathways on very specific points in specialized tissues (Costanzo-Garvey *et al*, 2009; Pearce *et al*, 2013; Rybakowska *et al*, 2014) (Appendix Fig S5C). Interaction with tissue-specific protein isoforms may thus cause highly cell-type-specific regulation by MAPKs.

The analysis of the best 100 hits for the NFAT4 class yielded results similar to the JIP1-type motifs, with some differences. In contrast, members of the greater MEF2A class were markedly dissimilar from those of the JIP1 class. Here, the proportion of cytoskeletal proteins was minimal, while the fraction of transcription factors was considerably higher. Proteins involved in other functions related to gene expression, such as chromatin remodeling or histone methylation, were also present in higher numbers. When comparing distributions of protein functions, the NFAT4 class appeared to lie between the two extremes represented by the JIP1 types (mostly cytoplasmatic targets) and greater MEF2A types (emphasizing nuclear actions) (Appendix Fig S6A). The similarity of NFAT4-type motif-containing proteins to JIP1-type bearing ones is easy to understand: Both primarily interact with JNK1. In certain protein families, one can discover closely related pairs in which one protein contains a JIP1-type docking motif and the other contains a likely independently evolved NFAT4-type docking motif (Appendix Fig S6B). On the other hand, the NFAT4-type motifs are structurally compatible with MEF2A types (unlike JIP1 types); thus, some of the predicted best binders are shared between the latter two lists. Our dot-blot experiments indeed corroborated that the overlapping motif definitions result in a naturally overlapping set of interactors for JNK1 and p38α (Appendix Fig S6C).

### Evolutionary analysis of D-motifs

MAPK pathways are found in almost every eukaryotic organism, and the three-tiered kinase cascade architecture of the MAPK module core is well conserved from yeast to human. Therefore, one would naturally expect the downstream targets of these pathways to be conserved as well. However, our results do not support this and in fact suggest the opposite. Although evolutionary conservation is considered to be a predictive feature of a functional linear motif, this did hold through for D-motifs. There was no correlation between FoldX (predicted binding energy) and any of the evolutionary conservation scores. The maximum traceable distance (MTD) of a motif in evolutionarily related species could be calculated from the eggNOG alignments. Here, we also noted that most of the motifs were traceable to vertebrates only. A more thorough search, using p-Blast searches in the UniProt database, revealed that some motifs are actually more ancient than what eggNOG data would suggest. Still, a high number of experimentally validated motifs were found to be relatively recent evolutionary inventions. After mapping the most distant organisms in which the motif in question is already present, we were able to compile an evolutionary histogram of MAPK-docking motif emergence (Fig 7A and Table EV3). Despite the fact that MAPK pathways are an eukaryotic common heritage, very few human docking motifs had an ancestry among unicellular organisms. This latter was only true for the MAPK kinases (MAP2Ks) or MAPK-activated kinases (MAPKAPKs) and a few truly ancient substrates, like MEF2/MADS-box proteins. Only in multicellular animals (Metazoa) did docking motifs become detectable in a variety of phosphatases and MAP3Ks as well as in the core set of mammalian substrates (ELKs, ATFs, JUNs, etc.). However, some of these motifs were difficult to find as they were subsequently lost in several lineages, especially in arthropods. The diversification of docking motifs continued in chordates, but it was in early vertebrates where a major re-wiring and expansion of MAPK partnerships occurred. Over 50% of the motifs identified in our experiments evolved at this period. After the development of bony fishes, motif emergence events became less common, but did not stop completely: New motifs appeared in lobe-finned fishes (Sarcopterygii), in terrestrial vertebrates (Tetrapoda), and even in mammals. Comparison of the known and predicted motifs from the best 100 hits for JIP1-type motifs suggests that there are many more recently evolved motifs in mammals (Fig 7B). These findings are well in line with recent results on yeast calcineurin interactomes: Yeast phosphatase-docking motifs were found to evolve fast, and their

---

**Figure 7. Evolutionary analysis of D-motifs.**

A   Analysis of MAPK-docking motif emergence (based on p-BLAST searches) paints a dynamic picture of MAPK pathway evolution. The panel was made based on 62 independent MAPK D-motif occurrences (see Table EV3). The histogram counts the number of known D-motifs (blue bars and blue numbers) and those newly identified in our experiments (red bars and red numbers). An approximate timeline is also added to give a realistic scale of the time dimension. The percentage of mammalian motifs found in selected model organisms is also indicated. Among model organisms, mice and zebrafish are relatively similar to human based on their MAPK interactomes. But fruit flies or yeast are rather poor models due to the low number of docking motifs being conserved across species.

B   Distribution of the 100 best JIP1-type motifs (predicted by the PSSM) versus their eggNOG-derived maximum traceable distance. Motifs validated experimentally as binders are represented under the green columns, while the total number of predicted motifs is shown in magenta. The analysis suggests that the most recently emerged motifs are still under-explored. While a reasonable percentage of D-motifs shared between humans and zebrafish were successfully validated in experiments, there also appears to be an intriguing number of (predicted) motifs restricted to mammals only. Note that this distance metric is different from the one used in (A) and extends to bony fishes only.

C   The branching pattern of closely related human proteins with MAPK-docking motifs points at rapid evolution. Most of the already-established MAPK partner proteins (blue bars) are members of families where more than one paralog carries the same motif. However, the more recently identified docking motifs (red bars) show a rather different picture. Proteins with stand-alone docking motifs thus appear to be much more common than previously expected.

D   Comparison of vertebrate and invertebrate genomes suggests that most novel D-motifs may provide paralog-specific regulation. The panel traces the emergence of D-motifs within protein families. Where multiple vertebrate paralogs carry the same motif, the docking elements are overwhelmingly pre-vertebrate inventions (often subject to motif loss, upper rows). However, where only a single paralog has the motif, the trend is exactly the opposite: Most motifs have typically newly evolved and have no counterpart in invertebrates (lower rows).

Data information: * denotes human paralogs containing validated D-motifs in (C) and (D).

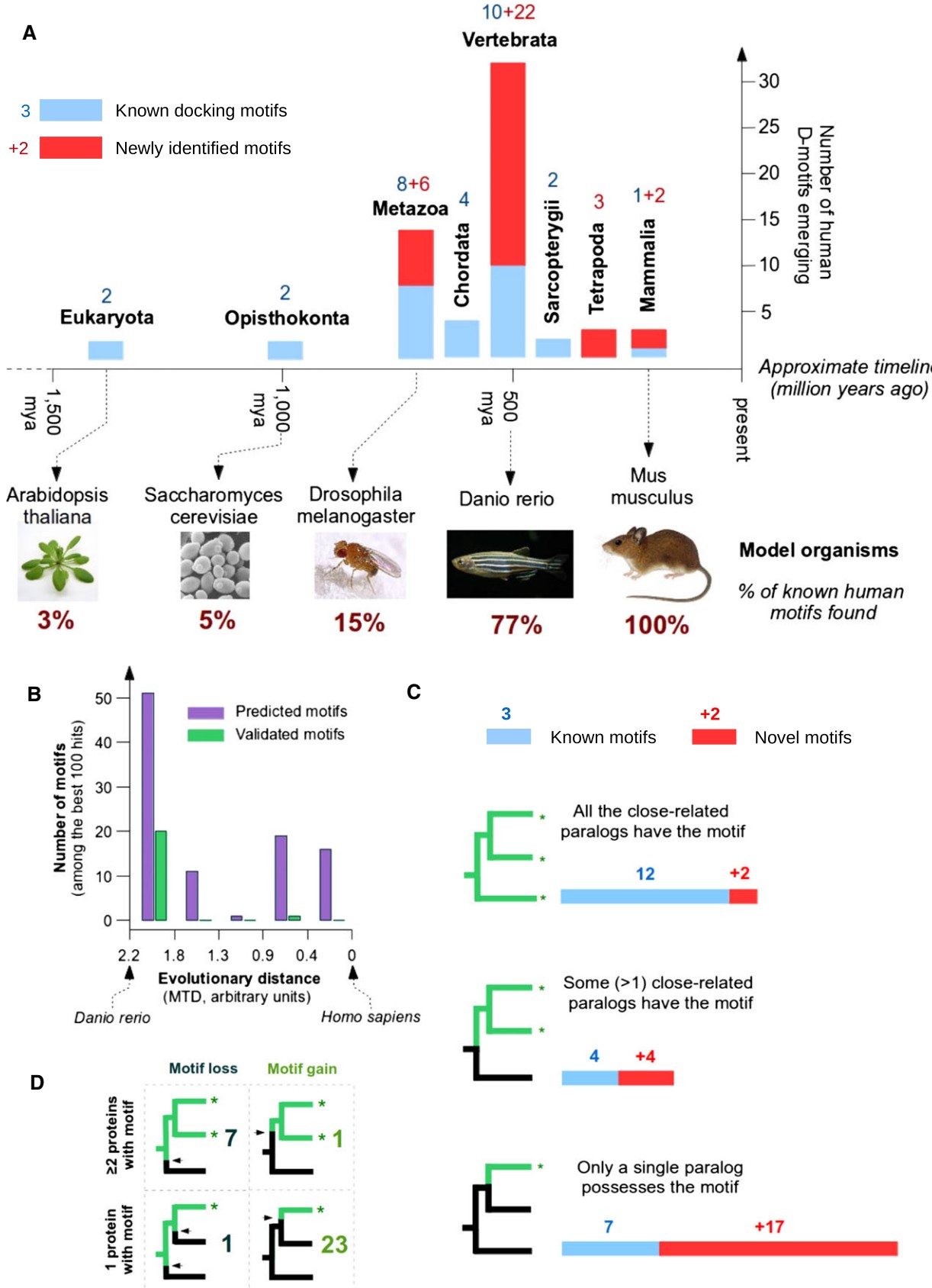

**Figure 7.**

distribution was highly divergent between related species (Goldman *et al*, 2014).

A further proof for the late evolution of MAPK partnerships is found when comparing paralogs (Fig 7C). These latter are closely related copies of the same ancestral gene that often preserved linear motifs from before their split. Most vertebrate proteins come in groups of 2, 3, or 4 closely related paralogs due to twin genome duplications—and subsequent gene loss—at the dawn of vertebrate evolution (Ohno, 1993; Durand, 2003). Interestingly, most of the better-known MAPK target proteins possess a D-motif in more than one vertebrate paralogs. However, the same is not true for the majority of novel partner proteins. Comparison of vertebrate proteins with those from earlier-branching genomes also helped us to determine whether a motif developed after the gene duplications or before (Fig 7D). Our analysis suggests that the presence of a motif in more than one paralog is predictive for ancient motif emergence. (Over 50% of such protein families have non-vertebrate members with the motif already in place.) In this case, motif loss appeared to be the dominant mechanism to create differences between vertebrate paralogs. Only a very few new motifs emerged in-between the two whole-genome duplication events, suggesting that this evolutionary stage was short-lived (Kuraku *et al*, 2009). On the other hand, where only a single paralog contained the motif, this motif was predominantly a new invention after the twin duplications—and not a result of an ancient motif being lost. This was the most common scenario for newly found D-motifs. Many of these novel MAPK-recruiting motifs are suspected to provide a paralog (or even isoform)-specific regulation, thereby offering unique roles to otherwise highly similar human proteins.

Having obtained a sufficient number of experimentally verified examples, we could also test some theories on the evolutionary processes creating the linear motifs. The motifs we validated (at least at a fragment level) could be classified based on their predicted origins (Fig 8 and Table EV3). Not surprisingly, the most common way of motif emergence appeared to be random mutations in an already-existing disordered segment. This can be illustrated by a known interactor, the Smoothelin-like protein 2 (SMTL2) (Gordon *et al*, 2013). Here, gradual sequence changes in terrestrial vertebrates led to the creation of the motif, which is restricted to placental mammals (Eutheria) (Appendix Fig S7A). There were also several examples for creation from scratch (i.e., from non-coding DNA). This could mean either translational start shift (translating an earlier non-translated 5′ UTR) or splicing site shift (leading to exonization of intronic sequences). While the N-terminal expansion of the protein is seen in MCL1 (as the motif-bearing segment has no counterpart in Bcl2 or in any other related protein), another newly identified partner, KSR2, serves as an example for splicing site rearrangements (Appendix Fig S7B). Here, the paralog KSR1 retains the ancestral intron–exon boundaries, which appear to have shifted in KSR2 (Appendix Fig S7C). We could even find examples for proteins where this mechanism may still be active: The motif can (in an isoform-specific way) be included or excluded due to alternative splicing or initiation. This is the case with the PDE4 genes, where most paralogs (PDE4A, PDE4B, PDE4D) still retain an ancestral, alternative exon containing a JIP1-type motif (Appendix Fig S7D).

Interestingly, linear motifs can also transmute into each other: Some examples in the dataset show potential switching between different MAPK-docking motif classes. As a result, distant organisms

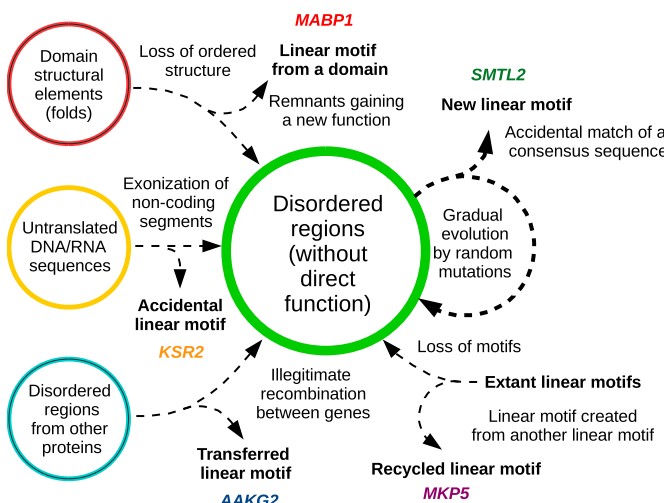

**Figure 8.  Mechanisms of binding motif emergence.**
D-motifs have diverse evolutionary origins. They most commonly originate from disordered protein segments via gradual accumulation of point mutations (e.g., SMTL2). However, they may also be created from folded domains (e.g., MABP1), or being introduced entirely *de novo*, from a previously untranslated genomic segment (e.g., KSR2). D-motifs can also be born from existing linear motifs, through gene fusion or recombination (e.g., AAKG2). But more commonly, this involves transmutation of a (previous) linear motif into a new one (example: MKP5). For more details, see Table EV3 and Appendix Fig S7.

may show different motif types at the same location: That is, the JIP1-type motif that we identified in MKP5 corresponds to an NFAT4-type motif in distant organisms (as in arthropods). In contrast, CCSER1 has an NFAT4-type motif in humans, but a JIP1-type one in zebrafish (Appendix Fig S7E). The motif in ELK1 is of the JIP1 type in humans, but Far1 type in protostomes. Also, the motif of TAB 1 is DCC-like in most non-vertebrate organisms, unlike the human which is MEF2A-like (Appendix Fig S7F). In some cases, a "horizontal motif transfer" (i.e., recombination between unrelated genes) may have complemented *de novo* emergence of motifs. This was the likely case for AAKG2, and the N-terminal region of AAKG2 (which is unique and sets it apart from AAKG1) showed a surprising similarity to the C-termini of MEF2A or MEF2C. In addition to the core motif, the disordered segments flanking the motifs also aligned well, and this cannot be explained by convergent evolution alone as the latter regions are not subject to the same selection (Appendix Fig S7G). The creation of a new linear motif from the unfolded remnants of earlier structured domains was yet another intriguing possibility. For the WDR62/MABP1 family, the duplication of WD40 repeats and their subsequent degeneration were the most likely source of the NFAT4-like D-motif (Appendix Fig S7H).

## Functional aspects of docking motif evolution

The typical purpose of docking motifs is to enable phosphorylation of recruited substrates (Reményi *et al*, 2006). Indeed, when the newly identified D-motifs from AAKG2 and DCX were mutated, their phosphorylation by their cognate MAPK was greatly reduced (Appendix Fig S8). Due to the lack of strict spatial constraints between D-motifs and phosphorylation sites, such roles can only be interpreted in the broader context of a protein. Unfortunately, most

phosphorylation target sites controlled by the novel docking motifs remain elusive. Yet in some cases, such sites either have been discovered beforehand or could be inferred based on spatial proximity and/or coevolution with D-motifs (Gordon *et al*, 2013). Examples for the latter give insight into how docking motifs emerged in relation to their target sites. In particular, both motif loss and gain are expected to have a profound effect on target sites: potentially endowing the protein with a new regulation. This is well illustrated by three cases: the nuclear factor of activated T cells (NFAT) family, with a *de novo* motif created in NFAT4 (adding on to a preexisting target site, Fig EV4A); the myocyte enhancer factor 2 (MEF2) family, displaying motif loss to a varying degree (with the concomitant loss of target sites, Fig EV4B); and the Grb2-associated binder (GAB) family, in which both events took place (Fig EV4C) (Chow *et al*, 1997; Yang *et al*, 1999; Wolf *et al*, 2015). To examine the evolutionary relationship between MAPK-binding D-motifs and phosphorylation target sites, the presence of validated D-motifs was analyzed in parallel with experimentally validated S/TP phosphorylation sites (Table EV4). This analysis was carried out in 50 proteins and compared the evolutionary conservation depth of the human motifs among vertebrate homologs (Appendix Fig S9). We found that there is a correlation in conservation of D-motifs and the most conserved putative phosphorylation sites, suggesting that D-motifs and putative target sites may have coevolved.

Our studies also support the notion that there are a number of proteins with multiple MAPK-docking elements. Apart from the case where these elements interact with different MAPKs, such as in the case of MKP5, where a rhodanese domain can bind p38 and a JIP1-type linear motif interacts with JNK, or GAB3, with separate motifs to recruit ERK1/2 or p38, the purpose of multiple D-motifs is unclear, especially because they tend to bind to the same surface and thus compete with each other for MAPK binding. Such domains or motifs are often not simple duplicates of each other and emerged independently during evolution. This happened in the case of BMPR2, where the first JIP1-type motif is found in almost all multicellular animals, but the second one is restricted to vertebrates (Podkowa *et al*, 2010) (Appendix Fig S10A). In addition, the ATF2/CREB family of transcription factors has an N-terminal motif next to a Zn-finger serving as the primary docking element in all metazoans, but several vertebrate paralogs have an additional JIP1-type motif with an unclear role (Appendix Fig S10B). As one MAPK molecule can only accommodate one motif at a time, it is probable that multiple docking motifs would allow several, mutually largely exclusive complexes—each with unique spatial orientation. As in the case of MKK7 which activates JNK1, the precise orientation of the MAPK versus the partner protein might have important implications on phosphorylating specific target sites (Ho *et al*, 2006; Kragelj *et al*, 2015).

D-motifs in proteins are known to facilitate phosphorylation of MAPK targets. However, the functional consequences of MAPK target site phosphorylation are unfortunately often not well understood. Phosphorylation may have diverse impact on protein function, and its relevance may only be revealed in the context of a signaling cascade. To this end, we have characterized the role of one of the newly found D-motif (527-RKVKPAPLEI-536) in the GAB1 signaling adapter protein (see Table 1 or Fig EV2, Far1-type) in HEK293 cell-based assays. GAB1 belongs to the insulin receptor substrate 1 (IRS1) family of adapter/scaffolding molecules playing a role in multiple signaling pathways (Holgado-Madruga *et al*, 1996).

Recently, it has been shown that cell membrane recruitment of GAB1 via its PH domain is controlled by an intramolecular switch (Wolf *et al*, 2015). The PIP3-binding surface of the GAB1 PH domain is masked by intramolecular interactions and phosphorylation at Ser551 unmasks this membrane binding surface and promotes recruitment of GAB1 to the cell membrane (Appendix Fig S11A). Interestingly, Ser551 in human GAB1 is a known MAPK S/TP target site, and we tested whether the newly found D-motif in this scaffold protein is indeed important for GAB1 membrane recruitment (Appendix Fig S11B), and whether it influences EGF/Ras→ERK2 signaling (Appendix Fig S11C). We found that docking motif versions of GAB1 had diminished capacity to translocate to the cell membrane and that these mutants were more sensitive to EGF stimulation regarding ERK2 activation. These results are fully consistent with GAB1's role as a complex regulator of EGF-mediated signaling: It exerts negative feedback control on the EGF/Ras→ERK2 pathway, presumably by relying on its ERK2-binding D-motif.

## Discussion

Protein kinases often use dedicated domains for substrate recognition. Known examples include the Src-family tyrosine kinases (SH2 and SH3 domains) (Alexandropoulos & Baltimore, 1996; Pellicena *et al*, 1998), SPAK/OSR kinases (unique domain) (Vitari *et al*, 2006) and Polo-like kinases (Polo-box domains) (Lee *et al*, 2014). In other cases, recruitment is provided by the catalytic domain itself, but by a distinct surface which is noncontiguous with the catalytic site. This appears to be common among relatives of MAPKs, the so-called CMGC (cyclin-dependent kinase/MAPK/glycogen synthase kinase 3/CDK-like) kinase group. However, each kinase family uses a different surface, with strikingly different recognition modes. Thus, motifs recognized by GSK3 or SRPK kinases (Dajani *et al*, 2003; Ngo *et al*, 2005) are unrelated to D-motifs or FxFP-motifs of MAPKs, or to CDK-docking motifs recognized by the cyclin subunit, for example (Lowe *et al*, 2002). Based on our results on MAPK-binding D-motifs, it may be anticipated that insights into other recruitment motif-based systems will greatly contribute to a system-level understanding of protein kinase-based intracellular signaling networks.

In the current study, it was demonstrated that canonical, D-motif-dependent partners of MAPKs are in fact quite common. However, a number of partners with atypical or "naturally defective" docking motifs do exist (e.g., MKK3, MKK7, TAB 1), and these are difficult to predict (Chang *et al*, 2002; De Nicola *et al*, 2013). Often such defective motifs act in a non-autonomous way: These weak elements may be complemented by additional protein stretches, motifs, or domains (Glatz *et al*, 2013). Besides, not all MAPK-binding elements are linear motifs. Folded domains such as the rhodanese domain of dual-specificity phosphatases may bind to the same site as intrinsically disordered docking motifs (Zhang *et al*, 2011). It should be noted that motifs other than the canonical D-motifs (e.g., the so-called FxFP-motifs) also exist (Jacobs *et al*, 1999; Fantz *et al*, 2001; Zhang *et al*, 2003). A considerable number of interactions might also be indirect, mediated by a third partner. Nevertheless, directly interacting with a MAPK solely through short linear motifs appears to be a major and widespread phenomenon in mammals. Although experimental testing of all putative MAPK D-motifs could not be performed, we suggest that the fraction of the

human proteome that harbors high-scoring D-motifs may be representative of the full interactomes for three distinct MAPKs, which may be best captured for JNK1 by the procedure presented in this study.

Some of the newly identified partners directly fit into the core of MAPK pathways. These include specific phosphatases as well as MAPK kinase kinases (MAP3Ks). While there can be little doubt that docking motifs of phosphatases would be required for MAPK dephosphorylation, the presence of docking motifs in MAP3Ks is a more intriguing observation. It is probable that phosphorylation of proteins acting on the MAP3K level (like on MEKK1, MLK1/2, or KSR2) would allow direct feedback control of MAPK pathways (Flotho *et al*, 2004). However, the majority of novel hits appear to lie outside the core MAPK pathway module, and these are probably simple downstream elements (i.e., substrates). Most of the novel proteins are expected to be either direct MAPK substrates or scaffold proteins (i.e., enabling phosphorylation of indirect MAPK substrates through protein complexes) (Fig EV5).

The wide distribution of D-motifs in a functionally diverse set of proteins explains how MAPKs can regulate such a broad spectrum of physiological processes. Previously, their specific regulatory roles were often attributed to single target proteins. For example, the role of JNK in axonal growth was attributed to the JNK–JIP1 interaction, and the association of JNK with diabetes was attempted to be explained by the JNK1–IRS1 interaction alone (Lee *et al*, 2003; Dajas-Bailador *et al*, 2008). In contrast, our results imply that these interactions may only be two examples of a substantially more complex protein network and JNK (as all MAPKs) connects to its targeted physiological systems by a large number of direct interactions. While individual connections might not be stable (especially in an evolutionary sense), multiple specific linkages could provide the key mechanism for a robust and adaptable physiological regulation.

Surprisingly, many of the newly implied MAPK partners have a restricted expression pattern enabling fine-tuned regulation in specialized tissues. Because of the latter phenomenon, a great deal of these interactions is unlikely to be discovered by large-scale protein–protein interaction screens. Easy-to-handle cell lines and mass-spectrometry-based analyses provide a powerful tool, but not for proteins that are only expressed in special, differentiated tissues (e.g., AAKG2, which is only abundant in cardiomyocytes) or restricted to certain embryonic developmental stages (e.g., DCX is almost exclusively expressed in developing neuroblasts) (Lang *et al*, 2000; Brown *et al*, 2003). Here, a modeling-driven interactome search is the most suitable tool to fill in the gaps in our knowledge. In addition, a reliable sequence-based prediction procedure sets the stage for easy examination on how MAPK signaling partners changed over time during evolution.

A comparative analysis suggests that rapid changes happened to MAPK pathways during the early evolution of vertebrates. Target proteins could have been brought under tight MAPK control by simply introducing docking motifs; however, this also necessitated target sites where phosphorylation could elicit functional effects. Similarly, existing links could have been thrown away by the loss of docking mechanisms. These processes were apparently the fastest when early vertebrates diverged from other chordates. Since then, the rate of motif emergence seemingly slowed down, but it has not completely stopped (some novel human motifs could be traced back only to mammals).

The current study explored MAPK–partner protein interactions in the human proteome. The identified D-motifs have the capacity to bind specific MAPKs *in vitro* and likely also in cells and in full protein context. Their functional relevance, however, remains largely unexplored. Undoubtedly, further studies will be required to address the relationship between these physical recruitment sites and MAPK phosphorylation target sites to understand how MAPK-mediated phosphorylation could elicit specific regulation. Nevertheless, this study suggests a rich and dramatically fast-evolving landscape for short recruitment sites and helps to explain how MAPKs could have become such widespread regulators of cellular physiology.

## Materials and Methods

### Motif scan and filtering by ANCHOR

Putative MAPK-binding D-motif instances were collected from the human proteome. Protein sequences were downloaded from the reviewed section of the UniProt database. The resulting Human Proteome Database (HPD) contains 20,248 sequences. The HPD was scanned for motif hits with basic pattern matching using the regular expressions of seven different D-motif classes/types (see Fig 1 and Appendix Fig S2), yielding 87,857 hits (JIP1-class, NFAT4-class, MEF2A-class-MEF2A-type, MEF2A-class-MKK6-type, DCC-class-generic, HePTP-class-Ste7-type, HePTP-class-HePTP-type). These hits were filtered using specific bioinformatics predictors aiming at selecting for biologically relevant instances. The filtering procedure followed a stepwise fashion as outlined below. Step 1: Filtering for the tendency to be part of disordered region that can undergo a disorder-to-order transition. The estimation of the interaction potential of the selected protein regions was done with the ANCHOR algorithm, a method trained to recognize binding regions in disordered protein segments (Mészáros *et al*, 2009). As described earlier, in linear motif selection, a more permissive version of ANCHOR can be used; therefore, the default 0.5 cutoff value was lowered in an adaptive way so that at least 90% of the known 47 formerly known D-motifs are retained (Mészáros *et al*, 2012). Motif hits were kept only if they overlapped with either an ANCHOR region predicted by using a 0.4 cutoff, or an ANCHOR region predicted by using a 0.3 cutoff, but in this case at least one of the 20 residue flanking regions of the motif hit had to have a sufficiently high average disorder value (above 0.45) predicted with IUPred (Dosztányi *et al*, 2005). As a result, the number of hits was reduced to 21,201. Step 2: Filtering for intracellular accessibility. Motif hits were discarded if they resided in proteins that were predicted to have a signal peptide by SignalP, and if they were also predicted not to have a transmembrane region predicted by Phobius (Käll *et al*, 2007; Petersen *et al*, 2011). These motif hits were predicted to be localized outside of the cell, which is incompatible with MAPK binding. Phobius alone was also allowed to predict signal peptides alone, if SignalP score were not too low (above 0.3). If a motif instance resided in a protein that was predicted to have a signal peptide but it also had at least one transmembrane region, the localization of the motif region was further checked. If it was entirely intracellular, it was kept, otherwise discarded. This filtering step reduced the number of motif hits to 18,952. Step 3: Filtering for correct localization. All hits that

were predicted by WoLF PSORT to be extracellular (with score $\geq 25$), membrane protein (with score $\geq 25$), localized to the E.R. (with score $\geq 15$) or the Golgi (with score $\geq 9$) were filtered out, unless they harbored transmembrane regions, and the region containing the motif was predicted to be localized in the intracellular space (Horton *et al*, 2007). There were 18,637 hits remaining after this step. Step 4: Filtering for structural accessibility. Motif hits that were determined to reside in Pfam domain regions were discarded (Finn *et al*, 2014). Some hits were also discarded in a manual curation process if they were located in Pfam Family/ Repeat/Motif regions likely to have a stable structure in isolation. Furthermore, motif occurrences that overlapped with coiled-coil regions predicted by COILS were removed as well (Lupas *et al*, 1991). Finally, there were 14,062 motifs remaining for further analysis including more than 90% of the known positives. Motifs passing all filters together with known positive hits are listed in Table EV1.

Compared to classical D-motifs, a "mirror image" like orientation for certain MAPK-binding motifs has also been described. In these "reverse docking" (revD) motifs, the hydrophobic stretch is located N-terminally to the charged residues. Apart from a short revD motif in PEA-15, all other mammalian examples are from a single group of MAPK interactors, the RSK/MAPKAPK family of kinases (Mace *et al*, 2013; Alexa *et al*, 2015). The low number of known reverse docking motifs, however, precluded their analysis in any systematic way.

### Scoring for structural compatibility by FoldX

Complexes of peptides with a MAPK were modeled using FoldX, similarly as it was previously described for SH2-binding peptides (Schymkowitz *et al*, 2005; Sánchez *et al*, 2008). For some motif classes, a reliable crystallographic model was available because the JNK1-pepJIP1 and JNK-pepNFAT4 complex crystal structures contained full consensus motifs (Heo *et al*, 2004; Garai *et al*, 2012). For p38α-pepMEF2A, p38α-pepMKK6, and ERK2-pepDCC crystal structures, the N-terminal positively charged amino acids binding in the CD groove were not located in the electron density, so for these cases the modeling was done only for the hydrophobic portion of the motif (Chang *et al*, 2002; Ma *et al*, 2010; Garai *et al*, 2012). The ERK2-pepHePTP complex contained artificial disulfide bond at the $\Phi_B$ position that facilitated crystallization of this protein–peptide complex; however, it also distorted the conformation of the docking peptide at the C-terminal region (Zhou *et al*, 2006). This was corrected by remodeling this part of the peptide by FlexPepDock. The model for the hypothetical Ste7 model was constructed from the yeast Fus3-pepSte7 and Fus3-pepMsg5 structures, superimposed on human ERK2, and then optimized with FlexPepDock (Reményi *et al*, 2005; London *et al*, 2011). List of the PDB files used to generate the various MAPK-docking motif structural templates are listed in Appendix Table S1. Each model contained the core motif with one extra alanine on both sides. The final FoldX score took into account the estimated energy of the complex as well as peptide stability in solution.

### Automated evolutionary sequence conservation analysis

All motifs were checked for evolutionary conservation. Sequence conservation, relative motif sequence conservation with respect to its flanking regions, and the maximum traceable evolutionary

distance of the motif (maximum distance between species bearing the same motif in closely related proteins) were all calculated from databases with pre-computed alignments. This was necessary to be able to compare the conservation of novel hits versus known motifs. For each protein with a potential motif, a cluster of orthologous proteins was extracted from the eggNOG database, using all vertebrates as the reference set of species (Jensen *et al*, 2008). Additionally, homologs from the inParanoid database were considered (Berglund *et al*, 2008). Here, the reference set of species consisted, in addition to human, of *Pan troglodytes*, *Mus musculus*, *Gallus gallus*, *Xenopus tropicalis*, *Danio rerio*, *Ciona intestinalis,* and *Brachiostoma floridae.* For each extracted cluster, only those sequences were retained that contained a motif instance within 10 or 50 amino acids compared to the human motif occurrence in full-length alignments in the eggNOG and inParanoid clusters, respectively. The regions containing the motif with its 10 amino acid flanks on either sides were retained and realigned using MUSCLE (Edgar, 2004).

### PSSM building, sequence logos, and final scoring

Position-specific scoring matrices for JIP1, NFAT4, greater MEF2A, and greater DCC classes were built including formerly known and newly found, validated human motif instances as well as all their identifiable vertebrate orthologs. To increase the sequence space coverage, we included more than just (known or novel) human MAPK-docking motifs. A method was devised to use evolutionarily weighted sequences for each independently evolved (or sufficiently unique) motif and to collect all known vertebrate orthologs. For this purpose, alignments were built from vertebrate proteins obtained by BLAST searches. Based on the refined consensus, motifs were classified as either potentially functional or non-functional. The motifs deemed potentially functional were realigned (with no gaps allowed) to the original sequence. In the end, the sequences were weighted by their evolutionary distance (based on the phylogenetic distances of the corresponding vertebrate organisms) and the final frequencies were obtained by summing up all independent groups with equal weights. In PSSM, each row represents one of 20 possible residues, and each column represents a position in a motif. Thus, the score for residue $X$ at position $i$ is defined in the following way:

$$X_i = \frac{\sum_s (w_s \times I(s_i = X)) + p \times X_b}{\sum_s w_s + p},$$

where $s$ is a peptide sequence, $w_s$ is the weight of that sequence based on the species from which it stems, $I$ is the indicator function which is 1 when its argument is true and 0 otherwise, $p$ is the pseudo-count defined as square root of total number of training peptides from the class and it is used to account for residues that do not appear at position $i$, and $X_b$ is the background frequency of the residue (based on UniProtKB/Swiss-Prot Release 2013.05). For computational efficiency and to account for background frequencies of residues, log-odds scores of $X_i$ were used in the form $X_i = \log (X_i/X_b)$. The final score was calculated as the sum of the log-odds scores of individual positions.

The distance between species A and B was the direct sum of branch lengths leading from A endpoint to B endpoint along the phylogenetic tree. These distances were the same that appear in

PhyloWidget visualization of Ensembl data (Jordan & Piel, 2008). In case of species not included there, evolutionary distances were extracted from trees published in literature and numerically re-calculated to fit the scaling of Ensembl-derived distances wherever possible (Barley *et al*, 2010; Agnarsson *et al*, 2011; Betancur-R *et al*, 2013). In the remaining few species where the tree topology was available without exact distance metrics, a numeric interpolation with equal weights was used.

Sequence logos were constructed using Seq2Logo-2.0 (Thomsen & Nielsen, 2012). Height of residue $X$ at position $i$ is directly proportional to its PSSM score $X_i$ (with $p = 0$) and $R_i$, information content of position $i$, defined as:

$$R_i = \log_2(20) - (H_i + e_n),$$

where $e_n$ is small sample correction parameter, expressed as $19/(2 * \ln(2) * \text{number of peptides})$, and $H_i$ is uncertainty at position $i$, defined as:

$$H_i = -\sum X_i \times \log_2(X_i).$$

Receiver operating characteristic (ROC) curve was constructed by adding simulated negative cases: Peptides in the human proteome conforming to the respective D-motif consensus but lying in a Pfam A-structured region were scored together with validated true-positive motifs, and at every value of the true-positive rate, the corresponding value of the false-positive rate was calculated and plotted (Finn *et al*, 2014). The area under the ROC (AUC) curve was calculated to assess the quality of the prediction: the closer it is to 1, the better is the predictor, and 0.9 indicates a very good predictor. For each motifs class, AUC calculation was performed in fivefold cross-validation setting with 100 samples, selecting 4 folds of D-motifs for training and 1 remaining fold, together with the D-motifs from Pfam A domains, for testing, while not allowing motifs from the same vertebrate orthology set to be used for both training and testing.

### Analysis on D-motif and phosphorylation site conservation

Consensus MAPK phosphorylation sites from proteins, which contain an experimentally verified D-motif (Table EV3), were collected from PhosphoSitePlus database (Li *et al*, 2002). Phosphorylation sites was selected if it was more than 10 amino acids away from the core D-motif and if there was site specific or mass-spectrometry experimental evidence of phosphorylation at that site in human. Normalized average traceable evolutionary distance for these sites from eggNOG vertebrate alignments has been estimated as the sum of the weights of the organisms where the phosphorylation site was present in the homologous protein divided by the sum of all the weights from the organisms in the alignment (Table EV4). If multiple candidate MAPK phosphorylation sites were present in the protein, the site with maximum normalized average traceable evolutionary distance was selected.

### Protein expression and purification

Activated ERK2, p38α, and JNK1 were produced by co-expressing the MAPKs with constitutively active forms of GST-tagged MAPK kinases (MKK1, MKK6, and MKK7, respectively) in *E. coli* (Rosetta pLysS). Full-length human MAPKs and their corresponding activator kinases were encoded in bicistronic vectors allowing the expression of hexa-histidine-tagged, phosphorylated MAPKs. Proteins were purified similar to dephosphorylated MAPKs as described earlier, using a Ni-column affinity and an ion-exchange step (Garai *et al*, 2012). The double-phosphorylated state of activated MAPK samples was confirmed by using anti-phospho MAPK antibodies with Western blots. Fragments corresponding to the D-motifs of various proteins (with appropriate flanks) were reverse-translated to synthetic DNA fragments endowed with sticky BamHI- and NotI-compatible ends. These were annealed, treated with PNK (polynucleotide kinase), and ligated into a customized pET-17 vector (pAZAD) containing flexible linkers (known to be intrinsically disordered) derived from ATF2 (Nagadoi *et al*, 1999). The linkers were incorporated directly between the GST-tag and the D-motif cassette as well as between the docking cassette and the target site. GST fusion constructs contained an ATF2-derived phosphorylatable epitope, lacking the T69 site, with only the T71 site present (ADQAPTPTRFL). All constructs were checked by DNA sequencing. The resulting GST-tagged artificial substrates with a C-terminal hexa-histidine-tag were expressed in *E. coli*, purified first on Ni-Sepharose and on glutathione-Sepharose by affinity chromatography. To prevent degradation of unstructured protein segments, lysates were treated with protease inhibitor cocktails (cOmplete EDTA-free inhibitor cocktail tablet, Roche), phenylmethanesuphonyl fluoride (PMSF) and benzamidine. All buffers following Ni-affinity purification contained PMSF (0.4 mM), benzamidine (2 mM), and EDTA (1 mM). Since detergents might influence immobilization efficiency, care was taken to use the same detergent, BOG (Octyl-β-D-glucopyranozide), in all final solutions (including dilutions) at the same concentration (0.02%).

### *In vitro* dot-blot phosphorylation assay

Stocks containing the purified artificial substrates were diluted to equal concentrations (~1 mg/ml), and printed (1 µl) as triplicates on nitrocellulose membrane using a pipetting robot (Hamilton STARlet). Membranes were dried at room temperature for at least 1 h. Prior to phosphorylation, membranes were blocked in Tris-buffered saline and Tween-20 buffer (TBS-T) containing 3% bovine serum albumin (BSA) for 30 min and washed three times with TBS-T. Phosphorylation was performed in kinase buffer containing 50 mM TRIS–HCl (pH = 7.5), 10 mM $MgCl_2$, 2 mM DTT, 0.1% BSA, and 2 mM ATP. Activated MAPKs were applied in 100–300 nM concentration. The phosphorylation solution was pre-mixed before applying onto the membrane. The reaction took place at room temperature on a rocker, it was stopped after 10 min by the addition of EDTA (at 25 mM end concentration), and membranes were then washed three times with TBS-T. Thereafter, membranes were blocked again by 3% BSA in TBS-T and developed by standard Western blot techniques using an anti-phospho-T71 ATF2 antibody (Cell Signaling Technology, #9221S) at 1:1,000 dilution and a secondary anti-rabbit antibody (Cell Signaling Technology, #7074S) at 1:2,000 dilution. After development with the Immobilon ECL kit (Millipore), phosphorylation signal was read either by luminescence (Alpha Innotech gel documentation system) or by fluorescence (Typhoon Trio+ scanner, GE). Non-phosphorylated membranes

were also checked for protein immobilization efficiency. The C-terminal hexa-histidine epitopes of GST phosphorylation reporter constructs were detected by a standard anti-His6 Western blot. Dot-blot experiments were performed for each construct at least twice, using different protein stocks. Only those constructs that consistently performed in all experiments above the non-cognate control were regarded as positive.

### *In vitro* protein–peptide binding assays

For fluorescence polarization (FP)-based binding affinity measurements, known MAPK-docking groove-binding peptides were N-terminally labeled with carboxyfluorescein or carboxytetramethylrhodamine (for ERK2, pepRSK1: PQLKPIESSILAQRRVRKLP STTL; for p38α, pepMK2: IKIKKIEDASNPLLLKRRKK or pepMEF2A: SRKPDLRVVIPPS; for JNK1, pep JIP3: RKERPTSLNFPL). 50 nM labeled reporter peptide was mixed with MAPKs in a concentration to achieve ~60–80% complex formation. Subsequently, increasing amounts of chemically synthesized test peptides were added, and the FP signal was measured with a Synergy H4 (BioTek Instruments) plate reader in 384-well plates. The Ki for each unlabeled peptide was determined by fitting the data to a competition binding equation. Titration experiments were carried out in triplicates, and the average FP signal was used for fitting the data with OriginPro7.

### Cell-based protein–protein interaction assay

The full-length cDNA of yellow fluorescent protein (YFP) was split at position 159, and fragments (F1 and F2) were pasted into pcDNA 3.1 vectors (Invitrogen). ERK2 was expressed as C-terminal and p38α and JNK1 as N-terminal F2 fusions. To facilitate expression, JNK1 and p38α had kinase-inactivating mutations (K55R and K53R, respectively), while ERK2 was wild type. The ERK2 and JNK1 constructs contained a FLAG-tag, but similarly tagged p38α constructs could not be expressed to a comparable extent. Therefore, expression levels of F2-p38α could only be monitored by an anti-p38α antibody. MAPK partners were expressed as N-terminal and C-terminal F1 fusions with FLAG-tags. F1 and F2 fusion pairs that gave the highest BiFc signal with wild-type MAPK partners were chosen to analyze the impact of docking motif truncations or mutations. These were introduced into full-length MAPK partner constructs by PCR or by the QuickChange method. N-terminal truncations were made in proteins with N-terminal docking motifs: MKK1(14-393), MKK6(18-334), RHDF1(26-855), MKP5(28-482), and FAM122A (14-287) and a 20-amino-acid-long C-terminal truncation was used to generate the DCX(1-343) construct. Mutation of multiple residues was utilized for proteins with internal motifs: JIP1 (R160A,P161A,L164A,L166A) and APBA2(R280A,P281A,L284A, L286A), and K331, K332, K333, and L337 were mutated to alanines in the short KSR2(325-399) construct All sequences were verified by DNA sequencing. HEK293T cells were cultured in Dulbecco's modified Eagle's medium (DMEM, Lonza) containing 10% fetal bovine serum and 1% penicillin/streptomycin at 37°C in an atmosphere of 5% $CO_2$ in 25-cm$^2$ tissue culture flasks (Orange Scientific). Cells were seeded onto 96-well plate (tissue culture test plate 96F, TPP) at 60–70% confluence 24 h prior to transfection. The medium was changed to serum-free OPTI-MEM (Gibco). Transient transfections with Lipofectamine 2000 reagent (Invitrogen) were carried out

according to the manufacturer's instruction. Cells were assayed 2 days post-transfection. For BiFc signal intensity measurements, cells were washed and suspended in 100 μl PBS. Twenty microliters of this cell suspension (~20,000 cells) was aliquoted into a 384-well black-sided plate. Fluorescence intensity per well was measured using a Synergy H4 (BioTek Instruments) fluorescence plate reader (excitation/emission wavelength was 515/535 nm). Cells from 50 μl of the PBS suspension were collected, and samples were subjected to Western blots using anti-FLAG-tag antibody (Sigma, F1804). For imaging, transfected cells were examined with an Olympus IX81 microscope using an Olympus FluoView 500 confocal laser scanning microscope system (Hamburg, Germany). YFP fluorescence was imaged using 514-nm excitation and a 535- to 560-nm emission filter.

### Cell-based assays for EGF stimulation and monitoring GAB1 localization

GAB1 constructs were subcloned into modified pCerulean-C1 vector with N-terminal Cerulean fluorescent protein and C-terminal FLAG fusion tags. HEK293T cells were cultured and transfected similarly as described above. Cells were transfected with 0.2 μg GAB1 DNA constructs and were serum-starved for 24 h. The media were removed after 40 h from DNA transfection and 100 μl PBS was added to wells. ERK pathway stimulation was started by the addition of epidermal growth factor (EGF, Sigma, E9644) at 20 ng/ml concentration to each well, and stimulation was terminated at different time points by adding 35 μl of 4× SDS loading buffer to wells. Cells were lysed and 10 μl of each sample was subjected to SDS–PAGE and transferred to PVDF membrane (Sigma, P2563). Western blotting for monitoring total ERK1/2 and GAB1 was done by using anti-p42/44 MAPK (Cell Signaling, #4695) and anti-FLAG antibody (Sigma, F1804), respectively. After stripping the membrane, phospho-p44/42 MAPK (ERK1/2) (Thr202/Tyr204) antibody (Cell Signaling, #9101) was used to detect ppERK1/2 protein levels. For secondary antibody, anti-rabbit HRP-linked antibody (Cell Signaling, #7074) and anti-mouse HRP-linked antibody (Calbiochem, #401215) were used. The ECL signal was quantified with an Alpha Innotech Fluorchem FC2 instrument.

For visualizing GAB1 localization analysis, transfected cells were examined by Zeiss LSM 710 confocal laser scanning microscope using 405-nm laser for excitation in chambered borosilicate cover glass (Lab-Tek, #155411). Evaluation of CFP-GAB1 localization was done using ImageJ. Cells were treated by 100 ng/ml epidermal growth factor (EGF) for 10 min.

### *In vitro* kinase assays

Activated JNK and p38α were produced by co-expressing the MAPKs with constitutively active forms of GST-tagged MAPK kinases (MKK7 and MKK6, respectively) in *E. coli* with bicistronic modified pET vectors. DCX and AAKG2 proteins and their mutants were produced in *E. coli* using modified pET vectors with C-terminal His-tag and N-terminal MBP-tag or GST-tag, respectively. Proteins were purified with double-affinity chromatography using Ni and MPB or GST column steps.

Twenty nM activated MAPK was incubated with 400 nM DCX or AAKG2 proteins at room temperature in the presence of 1 mg/ml

BSA. Kinase reactions were carried out in 50 mM HEPES pH 7.5, 100 mM NaCl, 5 mM MgCl$_2$, 0.05% IGEPAL, 5% glycerol, 2 mM DTT in the presence of 250 μM ATP, and ~5 μCi of ATP (γ32P). Reactions were stopped with protein loading sample buffer complemented with 100 mM EDTA, boiled, and then subjected to SDS–PAGE. The dried gel was analyzed by phosphorimaging on a Typhoon Trio+ scanner (GE Healthcare). Competitor docking motif peptides were chemically synthesized and used in 10 μM concentration (pepMK2, specific to p38: IKIKKIEADASNPLLLKRRKK; and pepJIP1, specific to JNK: DTYRPKRPTTLNLFP).

Expanded View for this article is available online.

## Acknowledgements

We are grateful to András Patthy and Ádám Póti for excellent quality chemically synthesized peptides, to Marianna Rakács and Gergő Gógl for their help in protein production, to László Végner for help in operating the pipetting robot for dot-blot assays, and to Bálint Szeder for help with confocal laser scanning microscopy. A.R. is the recipient of the "Lendület" Grant from the Hungarian Academy of Sciences (LP2013-57). This work was also supported by the OTKA NN 114309 grant (awarded to A.R.). Z. D. acknowledges the support of the "Lendület" Grant from the Hungarian Academy of Sciences (LP2014-18) and OTKA grant (K108798) B.M. acknowledges the support from OTKA grant NK 100482.

## Author contributions

AZ and AR designed the study. AZ, AA, ÁG, and KK performed experimental work. TB, BM, and OVK performed *in silico* analysis. AZ, OVK, ZD, and AR analyzed data. AZ and AR wrote the paper. All authors read and approved the manuscript.

## Conflict of interest

The authors declare that they have no conflict of interest.

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
