## [Review Process File · Molecular Systems Biology]

Systematic discovery of linear binding motifs targeting an ancient protein interaction surface on MAP kinases

András Zeke, Tomas Bastys, Anita Alexa, Ágnes Garai, Bálint Mészáros, Klára Kirsch, Zsuzsanna Dosztányi, Olga V. Kalinina and Attila Reményi

Corresponding author: Attila Reményi, HAS Research Centre for Natural Sciences

Review timeline:

Submission date:	29 April 2015
Editorial Decision:	09 July 2015
Revision received:	15 September 2015
Editorial Decision:	25 September 2015
Accepted:	01 October 2015

Editor: Thomas Lemberger

Transaction Report:

1st Editorial Decision

09 July 2015

Thank you again for submitting your work to Molecular Systems Biology. First of all, I would like to apologize for the delay in getting back to you, which was due to the late arrival of the report of Referee #1. We have now finally heard back from the two referees who agreed to evaluate your manuscript. As you will see from the reports below, the referees find the topic of your study of potential interest. Referee #2 raises however concerns on your work, which should be convincingly addressed in a revision of this work. One of the main issues refers to the physiological significance of the novel interaction, in particular with regard to MAPK-dependent phosphorylation in a cellular context.

If you feel you can satisfactorily deal with these points and those listed by the referees, you may wish to submit a revised version of your manuscript. Please attach a covering letter giving details of the way in which you have handled each of the points raised by the referees. A revised manuscript will be once again subject to review and you probably understand that we can give you no guarantee at this stage that the eventual outcome will be favorable.

Reviewer #1:

This study of linear binding motifs which control MAPK signalling is not only timely, it is comprehensive and very well carried out.

The paper should be scheduled for imminent publication. MSB is lucky to receive this study, it could easily be considered for NATURE BIOTECH or MOLECULAR CELL and alike as well in the reviewers view point.

The study of MAPK signalling is very challenging and this study goes to great extend using a plethora of methods to ID and validate new binding/docking motifs. The paper furthermore explores the evolutionary aspects of this exciting biological system and validates the role of several motifs in the context of cell biology.

The only critique would be that if the authors had added more cell/cancer/other biology the publication could probably fly even higher in terms of impact ceiling but of course it is also critical to get out as it will function as a resource for the community.

I am happy to give minor critical/technical suggestions in a revision but it should be as 'accepted with minor revisions'.

Reviewer #2:

This paper by Zeke et al. describes systematic discovery of linear docking motifs that bind to Mitogen-activated protein kinase (MAPK) by using combined computational and experimental approach, which was further studied in an evolutionary point of view of MAPK networks. PPIs mediated by short linear motifs with low affinity and disordered structure are generally less well understood due to limitation of current PPI screening methods. Using a combination of string matching, filtering with ANCHOR/Pfam and FoldX-based energy calculations, the authors then identified potential motifs to validate in vitro and cell-based system, enabling identification of novel MAPK docking motifs. From the selected novel motifs, it was suggested that most human MAPK binding motifs are newly emerged in MAPK pathway evolution. Linear motifs in general, and in particular interactions mediated by kinase docking motifs are generally understudied compared to other interactions, hence this work is relevant and topical. However, a number of competing approaches exist to detect interactions mediated by linear binding motifs to specific binding domains. In particular, the approach of computationally filtering/selecting candidate motifs for subsequent assays by experimental validation is reminiscent of earlier work (e.g., by Cesareni and co-workers). Nevertheless, the authors designed a clever system specifically targeted towards D motifs and present a compelling story complete with in vivo validations and successfully selected novel D-motifs and expand the MAPK interactome.

I have a few comments on this paper,

- (1) The FoldX score was supposedly used to rank the putative motif instances to be tested experimentally. When I look at Table S2, I see tested motifs all over the place, without any pattern that was obvious to me. I re-read the appropriate sections in the manuscript multiple times, but could not figure out the actual procedure.
- (2) As a (relatively weak) correlation of the FoldX results with the later PSSM scores appears to exist, why not use the FoldX energies as part of a combined scoring function? It appears to me that there is money left on the table here.
- (3) In the evolutionary analysis, why not analyze the docking motifs in parallel with the actual MAPK phosphorylation motifs and look for differences/similarities?
- (4) The authors only found and tried to confirm the novel interactions, but never traced real cellular consequences (phosphorylation) or mechanism in which the interaction is involved. How does the binding of MAPK to putative D-motif guarantee the real relevance in the context of MAPK signaling cascade? In other words, the authors may confirm MAPK-dependent phosphorylation of native target proteins in vitro or in vivo with D-motif lacking versions as control.
- (5) The authors confirmed bindings between motif-source protein and MAPKs by cell-based YFP fragment complementation. But the results in cellular context seems to be inconsistent, especially in case of p38a. Is this because of weak interaction?
- (6) It may be good to point out how many kinases (e.g., in human) are thought to have docking sites (and thus interact with D-motifs). As far as I recall it is not just MAPKs, so the approach could presumably be extended to all kinases with a docking site.

(7) Need proper numbering of figure legend for Fig. 3.

To me this is a nice story that encompasses a range of different analyses and techniques from structural modeling all the way to in vivo validation and evolutionary analysis. Of course (and this may be relevant especially to MSB), it is a localized (to one, albeit important, pathway) analysis, rather than a whole systems view...

In either case, I think the above comments do need to be addressed before publication

1st Revision - authors' response

15 September 2015

Reviewer 1:

Naturally, we greatly appreciate the overwhelmingly positive and encouraging comments of the Reviewer.

Reviewer 2:

The major comments regarding additional experimental validation for the relevance of newly identified D-motifs as well as the idea to carry out an analysis on MAPK D-motif and target motif conservation in parallel became part of the revised manuscript (see in the last section of the Results). For all the comments please find our answers below.

(Comment 1 - selection of motifs to be tested)

The FoldX score was supposedly used to rank the putative motif instances to be tested experimentally. When I look at Table S2, I see tested motifs all over the place, without any pattern that was obvious to me. I re-read the appropriate sections in the manuscript multiple times, but could not figure out the actual procedure.

Acknowledgedly, the chosen motifs for experimental tests may indeed look somewhat haphazardly chosen. Nevertheless, the absolute majority of tested motifs are still among the best 25% or 50% of motifs (as scored by FoldX), but also include some lower-ranking hits.

The selection of hits for testing did not only take the FoldX scores into account, but also their evolutionary conservation. The reason for this was two-fold: We wanted to describe novel motifs with biologically important functions (highly conserved motifs may have a higher chance of being biologically critical) and also be able to trace their evolutionary history and origins (this was also important for the generation of evolutionarily weighted PSSM scores). Since the majority of already-described motifs are highly conserved, we even experimented with combining evolutionary conservation and FoldX scores into a single score. However, given the lack of correlation between these scores and the tendency of certain, otherwise perfectly functional D-motifs to lack conservation, this did not live up to our first expectations. Thus, Figure 3 (panel B) highlights the evolutionary depth bias incurred by selecting the more conserved motifs for testing. But the resulting PSSM scores (and the "best 100" lists) still give an unbiased estimate of motif goodness and evolutionary depth, as they are based on sequence similarity to a large set of experimentally validated, evolutionarily unrelated motifs.

(Comment 2 – FoldX energies as part of a combined scoring function)

As a (relatively weak) correlation of the FoldX results with the later PSSM scores appears to exist, why not use the FoldX energies as part of a combined scoring function? It appears to me that there is money left on the table here.

Structure based exploration of linear motif space naturally lags behind that of sequence based analysis. Nevertheless, the former may greatly add to the latter provided that reliable structural models are available for protein-peptide complexes. For many docking motif based systems (see Comment 6), this is not available. In addition, FoldX energies show correlation to final PSSM scores depending on the motif class, indicating that the available MAPK-docking peptide complex models could vary in their capacity to represent valid binding motif conformations. We would argue that currently a PSSM based search algorithm - where FoldX energies aids the construction of a reliable sequence based approach – is the pragmatic way to go. Future studies, which will attempt to use more and more structure based approaches will be required to answer to what extent protein-peptide structural models will be useful to explore functional binding sites on the systems level.

(Comment 3 – evolutionary analysis on docking motifs and phosphorylation motifs)

In the evolutionary analysis, why not analyze the docking motifs in parallel with the actual MAPK phosphorylation motifs and look for differences/similarities?

An evolutionary sequence conservation analysis on MAPK target motifs and D-motifs became part of the revised manuscript (see **Figure S10** and **Table S6**). Here we analyzed target motif conservation parallel to that of D-motifs from proteins that contain experimentally validated docking motif instances. In brief, this analysis on 50 MAPK binding D-motifs and phosphorylation target site pairs suggests that these functionally coupled sites have likely co-evolved, as their sequence conservation among vertebrate orthologs seem to be well-correlated.

(Comment 4 – relevance of D-motifs in native target proteins)

The authors only found and tried to confirm the novel interactions, but never traced real cellular consequences (phosphorylation) or mechanism in which the interaction is involved. How does the binding of MAPK to putative D-motif guarantee the real relevance in the context of MAPK signaling cascade? In other words, the authors may confirm MAPK-dependent phosphorylation of native target proteins in vitro or in vivo with D-motif lacking versions as control.

We analyzed the impact of D-motif ablation on native MAPK target proteins (AAKG2 and DCX) in vitro (see **Figure S8**). In addition we also elucidated the role of the D-motif in GAB1 in cell-based assays, in the context of a MAPK signaling cascade (see **Figure S12**). All three D-motifs were identified first in our study, and results of our new experiments fully support the relevance of D-motifs on p38a mediated phosphorylation of AAKG2, JNK1 mediated phosphorylation of DCX, and ERK2 mediated phosphorylation and complex regulation of GAB1.

(Comment 5 - differences between ERK2 and p38 α in BiFC experiments)

The authors confirmed bindings between motif-source protein and MAPKs by cell-based YFP fragment complementation. But the results in cellular context seems to be inconsistent, especially in case of p38a. Is this because of weak interaction?

Part of the differences seen between ERK2 and p38a in the cell-based experiments (using full-length proteins) might stem from interactions outside the D-motif. It is true that the D-recruitment sites of ERK2 and p38a are overwhelmingly similar. However, their other binding sites, such as the FxFP site may be dissimilar. There is a marked difference between preferences of FxFP-type ligands between ERK1/2 and p38a/ β kinases for example.

Another part of the differences might stem from the experimental setting itself.

Unfortunately, we were never able to express FLAG-tagged p38a to an extent comparable to ERK2. While removal of the FLAG-tag improved expression of p38a, this simultaneously made it impossible to directly compare its exact levels to ERK2 (as they were now recognized by two different antibodies). Therefore the comparison of raw fluorescence levels between the ERK2 and p38a constructs is usually not informative: One should look at the relative reduction seen in the D-motif-less mutants versus the wild-type proteins. Additionally, due to the nature of bimolecular fluorescent fragment complementation assay, negative results are not informative. We only regarded those interactions to be relevant where fluorescence of the wild-type protein was above the noise threshold (different for each construct), and the reduction after the removal of the D-motif was significant (therefore weaker or sterically unfavorable interactions might have ended up being undetected under these conditions). Our dependence on the threshold can explain why interactions with all three MAPKs were detected with DCX (where the noise background was close to zero), but we failed to see an interaction between p38a and KSR2 (where the background noise was relatively high, compared to signal strength).

(Comment 6 - docking motifs for other protein kinases)

It may be good to point out how many kinases (e.g., in human) are thought to have docking sites (and thus interact with D-motifs). As far as I recall it is not just MAPKs, so the approach could presumably be extended to all kinases with a docking site.

A brief summary of protein kinases using docking motif based recruitment is now included into the Discussion (see first paragraph).

We indeed believe that a similar approach that we implemented on MAPKs in this study could be also used to explore other docking motif based systems in the future. However, there are some caveats. While the X-ray structures of several other kinase - docking motif complexes have also been determined, in most cases this means a single structure for an entire family of motifs. Unfortunately, our method critically relies on the existence of multiple, experimentally determined

structures to be able to set up structurally homogenous classes. And that is rarely satisfied with most kinases, save perhaps the cyclin-dependent kinase CDK2, where multiple cyclin A - partner motif complexes are available. Interestingly, the latter case also suggests that CDK2 / cyclin A docking motifs are structurally not homogenous, and must be split into at least two structural subclasses $((\theta/\varphi-\theta)-\theta-x-\varphi_1-x-\varphi_2$ or $(\theta/\varphi-\theta)-\theta-x-\varphi_1-\varphi_2$ -Gly in our symbolic notation, with θ being positively charged and φ being hydrophobic amino acids. This would allow the extension of the core "RxL" consensus in a meaningful way. But even there, the number of known examples is too low to allow drawing firm conclusions. A systematic study of these motifs will only be possible once our knowledge expands enough so that a set of a dozen or so, evolutionarily independent and experimentally proven examples become available.

(Comment 7 – Correcting Fig 3. legend panel labeling)

Need proper numbering of figure legend for Fig. 3.

This was fixed by correcting the figure legend.

2nd Editorial Decision

25 September 2015

Thank you again for submitting your work to Molecular Systems Biology. We have are now globally satisfied with the modifications made and we will be able to accept you manuscript for publication pending a few minor points.
